# The Large-eddy Observatory - Voitsumra Experiment 2019 (LOVE19) with high-resolution, spatially-distributed observations of air temperature, wind speed, and wind direction from fiber-optic distributed sensing, towers, and ground-based remote sensing

Karl Lapo[1,2], Anita Freundorfer[1], Antonia Fritz[1,3], Johann Schneider[1], Johannes Olesch[1], Wolfgang Babel[1,2], and Christoph K. Thomas[1,2]

[1]University of Bayreuth, Micrometeorology Group, Bayreuth, Germany
[2]Bayreuth Center of Ecology and Environmental Research, BayCEER, Bayreuth, Germany
[3]*now at*: University of Innsbruck, Innsbruck, Austria

**Correspondence:** Karl Lapo (karl.lapo@uni-bayreuth.de)

**Abstract.**

The weak-wind Stable Boundary Layer (wwSBL) is poorly described by theory and breaks basic assumptions necessary for observations of turbulence. Understanding the wwSBL requires distributed observations capable of separating between submeso and turbulent scales. To this end, we present the Large Eddy Observatory, Voitsumra Experiment 2019 (LOVE19) which featured 2105 m of fiber optic distributed sensing (FODS) of air temperature and wind speed, as well as an experimental wind direction method, at scales as fine as 1 s and 0.127 m in addition to a suite of point observations of turbulence and ground-based remote sensing profiling. Additionally, flights with a fiber optic cable attached to a tethered balloon provide an unprecedented detailed view of the boundary layer structure with a resolution of 0.254 m and 10 s between 1 to 200 m height. Two examples are provided demonstrating the unique capabilities of the LOVE19 data for examining boundary layer processes: 1) FODS observations between 1 m and 200 m height during a period of gravity waves propagating across the entire boundary layer and 2) tracking a near-surface, transient submeso structure that causes an intermittent burst of turbulence. All data can be accessed at Zenodo through the DOI 10.5281/zenodo.4312975 (Lapo et al., 2020a).

## 1 Introduction

The lowest portion of the atmospheric boundary layer, coined the "critical zone" (Brantley et al., 2007), is where heat, water vapor, carbon dioxide, pollutants, among other constituents are exchanged and mixed between the atmosphere, biosphere, and hydrosphere. As such, this thin layer of the atmosphere and its coupling to the surface plays an important role for humans and ecosystems. However, there is a disparity between the actual boundary layer and the theoretical understanding during periods

with weaker winds and statically stable conditions, which we refer to as the weak-wind Stable Boundary Layer (wwSBL).

This categorization of the SBL largely overlaps with the other categorizations, such as very stable boundary layer (vSBL). During the wwSBL, turbulence can range from small, but finite mixing to large, intermittent bursts of turbulence. In both cases turbulence does not have a strong relationship to local variables and is therefore poorly described by similarity theory (Sun et al., 2012, 2020; Pfister et al., 2021a). While some aspects of the transition between these two states are understood (e.g., Van de Wiel et al., 2017) the drivers remain largely unknown (Abraham and Monahan, 2020; Acevedo et al., 2014; Mahrt et al.,

2020). Further complicating our understanding of the wwSBL, assumptions necessary to invoke when observing atmospheric turbulence, for instance Taylor's hypothesis of frozen turbulence when converting the average flux in time at a point to represent the area-averaged flux, are not valid (Mahrt, 2008; Mahrt et al., 2009; Sun et al., 2015; Pfister et al., 2021b, a).

A growing body of evidence suggests that submeso-scale atmospheric structures play a substantial role in the turbulence generation of the wwSBL and the subsequent disagreement between theory and observations. Submeso-scale atmospheric

structures are loosely defined as structures larger than turbulent eddies but shorter than the mesoscale, typically taken as greater than 10 m and up to kilometers with time scales between 20 s and an hour (Thomas, 2011; Thomas et al., 2012; Zeeman et al., 2015; Pfister et al., 2021b, a; Mahrt et al., 2009; Mahrt, 2010; Abraham and Monahan, 2020; Mahrt et al., 2020). These structures have been shown to include a diverse range of phenomena including internal waves (Sun et al., 2015; Cava et al., 2017, 2019; Petenko et al., 2019), horizontal wind direction meandering (Cava et al., 2017, 2019; Lang et al., 2018;

Mortarini et al., 2019), semi-stationary thermal submeso fronts (Mahrt, 2017; Pfister et al., 2021b, a; Kang et al., 2015), and transient cold-air motions (Thomas et al., 2012; Zeeman et al., 2015) often with these phenomena co-occurring. Generally, little is known about the specific properties and drivers of these structures since they occur at a gap in our observational capabilities, especially near the surface (see above citations). As these structures are hypothesized to play an important role in the physics of the wwSBL and understanding their characteristics, driving factors, and influence on turbulence is a major goal of boundary

layer research (Sun et al., 2015; Mahrt and Thomas, 2016).

Using the next generation of surface meteorological observational techniques that are specifically aimed at observing submeso-scale motions and their role in the wwSBL (Thomas et al., 2012; Zeeman et al., 2015), the European Research Council Horizons 2020 project "DarkMix" seeks to reveal this "dark side" of turbulence. One of the key needs for studying submeso-motions and their impact on turbulence in the wwSBL is spatially-distributed observations of atmospheric properties

with a fine enough spatial and temporal resolution to separate between the submeso and turbulent scales and with a large enough spatial extent to resolve submeso modes (e.g., Mahrt et al., 2009; Acevedo et al., 2014; Abraham and Monahan, 2020; Pfister et al., 2021b, a). To that end, the Large-eddy Observatory - Voitsumra Experiment 2019 (LOVE19) featured a large array of fiber-optic distributed temperature sensing (DTS). This technique uses the temperature-dependent Raman-backscatter from laser pulses transmitted along a fiber optic cable in order to resolve temperature at a fine spatial (as fine as 0.127 m) and

temporal (as fine as 1 s) resolution (Selker et al., 2006; Tyler et al., 2009). DTS can resolve air temperature at scales fine enough to resolve the difference between turbulent and submeso scales (Thomas et al., 2012; Zeeman et al., 2015; Peltola et al., 2021; Thomas and Selker, 2021; Fritz et al., 2021; Zeller et al., 2021). It has also been developed to observe additional atmospheric properties on a distributed basis such solar radiation (Petrides et al., 2011), dew point (Euser et al., 2014; Schilperoort et al.,

2018), wind speed (Sayde et al., 2015; van Ramshorst et al., 2020; Pfister et al., 2019; Zeller et al., 2021), and recently wind direction (Lapo et al., 2020b). We refer to this broader family of sensing techniques as Fiber Optic Distributed Sensing (FODS, Thomas and Selker, 2021).

The scientific goal of LOVE19 was to observe submeso-scale structures and their role in generating near-surface turbulence in the wwSBL. FODS observations were set-up in a configuration intended to capture submeso-scale structures as they flow across the study area in order to provide a spatial context for more typical boundary layer observations such as point and profile observations on towers and ground-based acoustic and light remote sensing. The experiment featured FODS air temperature, wind speed, and wind direction totalling 2105 m of fiber-optic observations. LOVE19 was coined a "Large Eddy Observatory" (LEO) as it observed spatially-distributed boundary layer properties at spatiotemporal scales similar to those represented by the Large Eddy Simulation (LES) technique.

LOVE19 was also intended to be a test bed for expanding various FODS observational capabilities, building towards the eventual goal of a spatially-continuous fully-3D turbulent flow sensing technique. These improvements include a first demonstration of the FODS wind direction technique in an environmental application and the second-ever FODS wind speed deployment, including improvements to the technique such as vertically-oriented fibers and paired-fibers with identical radiative properties. Further, a tethered balloon was used to deploy a fiber-optic cable 200 m vertically in order to connect between observations of the surface layer and the remote sensing of the upper boundary layer with uniquely high resolution, spatially-distributed observations of air temperature (section 4.3 Fritz et al., 2021).

The site description and layout, experimental description, and data availability are given in section 2. Ground-based remote sensing and flux observations are described in section 3. Details of FODS operating principles as well as a description of the FODS components deployed during LOVE19 are presented in section 4. FODS wind speed and wind direction methods are discussed, and the FODS wind speed are evaluated in section 5. Finally, examples intended to highlight the novelty and merit of the LOVE19 data set for the broader atmospheric sciences community are presented (section 6). The first example demonstrates the unique insights from FODS for studying internal gravity waves (section 6.1). The second example examines a near-surface, spatially discrete submeso-scale structure and its effect on turbulence (section 6.2).

## 2    The LOVE19 campaign

Instruments were deployed at the bottom of a broad valley in the Fichtelgebirge mountains in Germany (50.0906° N 11.8543° E; 624 m asl; Fig. 1). The valley, which stretches from the southwest to the northeast, is surrounded by ridges up to 200 m tall to the north and up to 400 m tall to the south (Fig. 1c). It is open to the northeast, while being bound by a shallow saddle to the west (Fig. 1b). The dominant wind direction is along the valley, across the long axis of the study site (Fig. 2a). The site is characterized by intense cold-air drainage and pooling, exceptionally calm nocturnal winds, strong static stability, and horizontal wind direction meandering (e.g., Mortarini et al., 2019).

The experimental site was an agriculturally used perennial grass field approximately 200 m by 300 m in size (Fig. 2a). The general land use in the valley was largely perennial short-statured grassland and agricultural fields as well as isolated forest

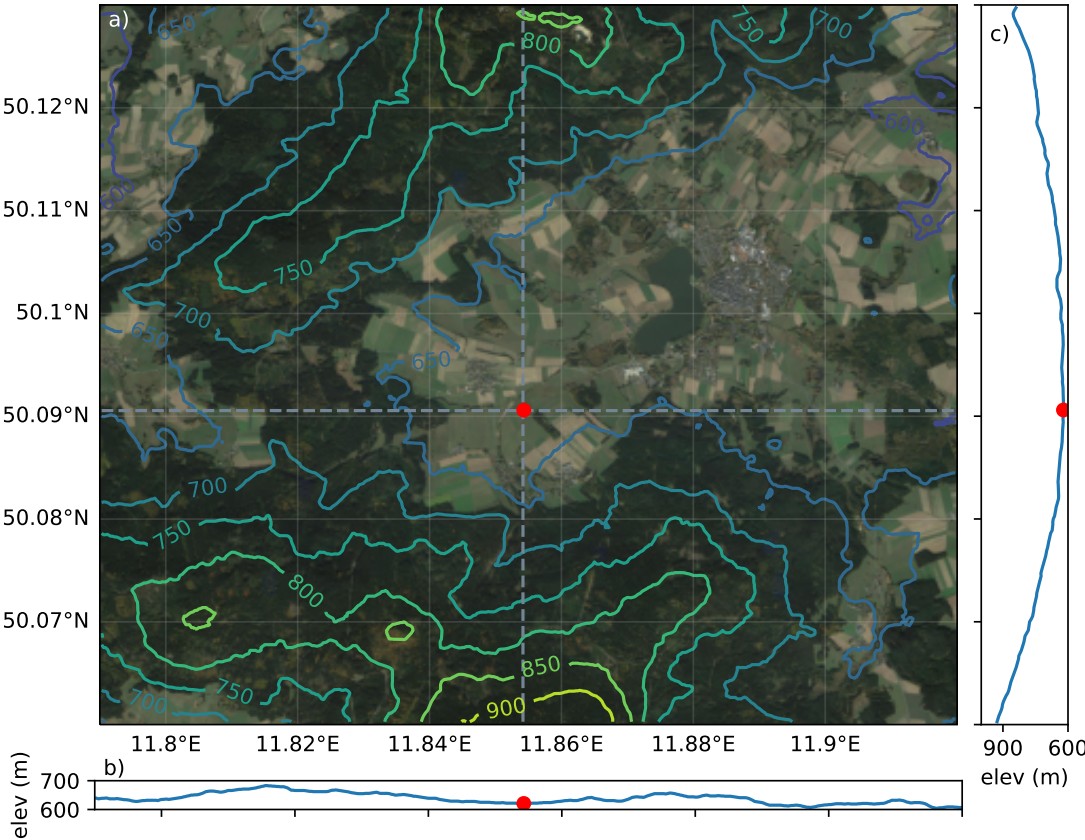

**Figure 1.** (a) The larger environment surrounding LOVE19 (red dot) with 50 m elevation contours from the Shuttle Radar Topography Mission (Farr et al., 2007). Dashed grey lines indicate the (b) the east-west and (c) north-south elevation transects. Map data © 2021 Google, map imagery © 2021 GeoBasis-DE/BKG, GeoContent, and Maxar Technologies.

patches. A few villages are scattered within a 5 km radius of the site. The site was situated at the bottom of a 40 m hill directly to the south. To the southeast lies an isolated patch of forest with a mean tree canopy height of 10 m while grass fields surround the field on all other sides, with a creek along the northern boundary.

Observations during LOVE19 were collected between June 6 to August 14. This period can be subdivided into three phases according the availability of the FODS components (Fig. 3). No FODS observations were collected throughout June and the first half of July, although all other components were operational. Between July 15 and July 28 observations from FODS with active-heating elements were collected, specifically in the form of a FODS-cross (Fig. 2a,d,e), distributed wind speed and air temperature observations along the outer rectangle (Fig. 2a,d), and distributed wind speed along the 12 m tower (orange and 95   gold components in Fig. 2a). On July 28 the active heating for FODS components was turned off decreasing the maintenance needs, and all FODS observations were switched from high-resolution to a ruggedized lower-resolution units (see section 4.1 for details). The number of days with significant precipitation ($\geq 1$ mm) was 5, 3, and 8 for the three phases, respectively.

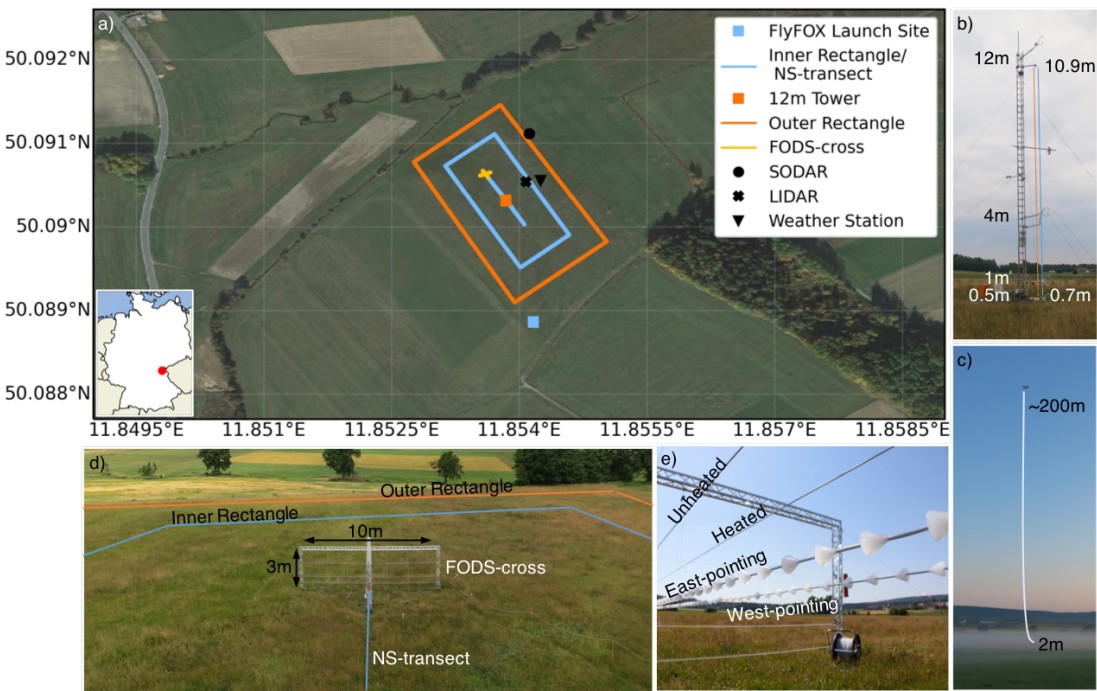

**Figure 2.** (a) A map of all components of the LOVE19 except for the ceilometer which was approximately 400 m to the northeast (not shown). FODS components with active heating for wind speed are shown in orange (outer rectangle and the 12 m tower), FODS components without heating are shown in blue (NS-transect and inner rectangle), and the FODS-cross is in gold. (b) The 12 m tower with CSATs and the paired heated-unheated fiber optic cables viewed from the east. (c) FlyFOX on July 18, a morning with fog discussed in section 6.1, viewed from the top of the hill to the south. (d) The outer rectangle with paired heated-unheated fiber optic cables (orange), the inner rectangle (blue), FODS-cross (aluminum trussing), and the NS-transect as viewed from the top of the 12 m tower. (e) The fiber quartet of the FODS-cross, consisting of paired heated-unheated and paired heated fibers with microstructures oriented in opposite directions are shown in more detail. To better highlight the fiber-optic cables they have been drawn onto images in b-d. Map data © 2021 Google, map imagery © 2021 GeoBasis-DE/BKG, GeoContent, and Maxar Technologies.

Significant rains primarily affected the data availability of the active-heating FODS elements during phase 2 as the moisture sometimes led to electric short-circuiting between the FODS cables and grass cover, which resulted in power loss due to the 100  false-current protection.

## 3   Ancillary observations

In addition to the FODS components, LOVE19 included more traditional boundary layer observations: point and profile observations of radiative and turbulent fluxes, air temperature, precipitation, and horizontal wind speed and direction, as well

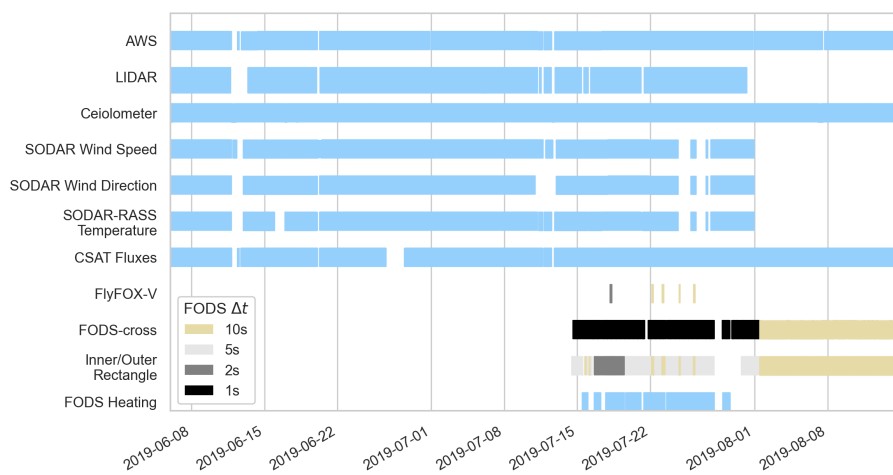

**Figure 3.** The data availability of the various observational systems during LOVE19. Specifically highlighted are the availability of the heating (necessary for FODS wind speed and direction) as well as the DTS instrument's temporal resolution, which varied with the campaign needs. Power failures occurred periodically due to electrical isolation issues heating the fiber optic cables sometimes co-occurring with rain events (section 7).

as ground-based remote sensing. All observations were made within 40 m of each other except for the ceilometer which was located approximately 400 m to the northeast of the other observations.

### 3.1 Ground-based Remote Sensing

The ground-based remote sensing combined Sound Detection And Ranging (SODAR-RASS), wind LIght Detection And Ranging (LIDAR), and a ceilometer. The SODAR-RASS (Model DSDPA90.64 and 1.29 GHz RASS, Metek GmbH, Elmshorn, Germany) measured the Doppler-quantities horizontal wind speed and direction, and sonic temperature with a 10 min temporal resolution, volume-integrated over 20 m vertical gates, in addition to non-Doppler acoustic backscatter intensity and vertical wind speed variance from spectral broadening (Thomas et al., 2006). The first physically meaningful observational gate center height was at 40 m above ground level (agl) and observations were available up to 300 m agl on average, while the maximum observation height varied with acoustic properties of the atmosphere and surrounding acoustic environment.

The Doppler wind LiDAR (Model Stream Line, Halo Photonics Ltd., Worcester, UK) was deployed in the middle of the field site to minimize near-surface flow distortion by obstruction. The LIDAR was employed to detect the height of the atmospheric boundary layer, obtain snap shots of the three dimensional wind field, and observe vertical wind speed. Consequently, the LIDAR was operated in three modes: a conical Vertical Azimuth Display (VAD) scan, Range Height Indicator (RHI) scan, and a vertical stare (VST). The vertical stare was employed for 14 minutes starting at each full hour with alternating VAD and RHI scans of 1 min duration between VST scans. The constant zenith angle VAD scans enabled retrieval of the horizontal wind speed and direction following the method of Browning and Wexler (1968) every 30 min at a zenith angle of 60° with

eight (every 45°) steps in the azimuth angle. The constant azimuth angle RHI scans were used to retrieve a snapshot of the cross-valley wind field every half an hour. RHI scans were performed along an azimuth angle of 327°, chosen because of the lack of obstacles, with 37 steps of 5° along the zenith. VSTs were performed for 14 min between the RHI and VAD scans, with a range gate of 24 m and a temporal resolution of 1 s. The VST data were aggregated to an 84 s time scale, which yields 10 averaging periods per vertical stare interval.

A ceilometer (Model CHM 8k, Lufft Mess- und Regeltechnik GmbH, Fellbach, Germany) was deployed to the northeast of the site as part of a new long-term flux observation site. The ceilometer measured the sky condition, cloud ceiling, total cloud coverage, and cloud penetration depths up to 8 km above the surface with a vertical resolution of 5 m.

## 3.2  Fluxes and ancillary observations

Four sonic anemometers (CSAT3, Campbell Scientific, Bremen, Germany) were located on the 12 m tower at heights of 0.5 m, 1.25 m, 4 m, and 12 m above the surface (Fig. 2a,b). Raw 20 Hz turbulence observations were processed with the flux eddy-covariance software 'bmmflux' (Thomas et al., 2009, see Fig. A1 for a schematic of the data flow). Fluxes and turbulent quantities were calculated using perturbation time scales of 1 min without any coordinate rotation and 10 min and 60 min with 3D rotation (Wilczak et al., 2001). The range of time scales enables using the sonic anemometer observations for different types of research questions, with the short 1 min scale specifically intended for use in separating between the turbulent- and submeso-scales following Mahrt and Thomas (2016).

An Automatic Weather Station (AWS) was situated 40 m to the east of the tower. Air temperature and humidity (Model HMP45A Thermohygrometer, Vaisala, Finland; radiation shielded and electrically aspirated), horizontal wind speed and direction (Model wind vane and cup anemometer, Theodor Friedrichs and Co, Germany), and aspirated four component radiation (Model CNR4 Net Radiometer, Kipp and Zonen, The Netherlands) were observed at a 2 m height. A soil temperature profile was observed at depths of 0.05 m, 0.25 m, and 0.5 m using platinum resistance (PT-100) temperature probes. Air temperature was also observed immediately above the surface at a height of 0.05 m agl using an unshielded PT-100 to record the minimum air temperature according to WMO standards. Finally, precipitation was observed at 1 m height (OTT Pluvio[2] - Weighing Rain Gauge, OTT HydroMet, Kempten, Germany). All observations were reported as 10 min block averages.

## 4  DTS

Raman-spectra DTS uses the wavelength shifted backscattered photons from a laser fired along a fiber-optic cable. Some of the backscattered photons have a higher and lower frequency, known as anti-Stokes and Stokes bands, with the ratio of these backscattered photons depending on the temperature of the fiber (e.g., Fig. 4a; Selker et al., 2006; Tyler et al., 2009). DTS can measure air temperature with sufficient temporal resolution to resolve turbulent fluxes (Thomas et al., 2012), turbulent third-order moments (Peltola et al., 2021), in addition to submeso-modes (Zeeman et al., 2015; Pfister et al., 2019, 2021b). In this experiment we employed a high-resolution Ultima DTS (ULTIMA DTS (5km variant), Silixa, Elstree, United Kingdom) capable of 1s and 0.127m resolution as well as a lower resolution, ruggedized Silixa XT (XT-DTS, Silixa, London, United

Kingdom) capable of 5s and 0.254m resolution. All DTS devices were located in a climate-controlled instrument trailer on the perimeter of the study area.

For scientific applications it is necessary to continuously calibrate the DTS output (Hausner et al., 2011; van de Giesen et al., 2012; des Tombe et al., 2020) i.e. to transition from backscattered light intensities to calibrated temperatures (Fig. 4a to 4b). For this reason the fiber-optic cable is run through reference sections with a known temperature. Typically, reference sections are water baths in which the fiber is loosely coiled, with PT-100s observing the water temperature. However, water baths are difficult to maintain for long environmental deployments so instead two novel solid state reference baths were employed.

Each solid state reference section consisted of a 20 kg cylinder of pure copper with 4 interlocking parts, which allowed for an internal groove around which the fiber was wrapped. The temperature of each copper cylinder was controlled thermoelectrically by Peltier elements to within $\pm$ 0.06 K and observed with up to two high-accuracy 4-wire platinum resistance (PT-100) thermometers embedded within the copper body next to the fiber-optic cables. The walls of the internal groove containing the fiber-optic cables was painted with a high-emmissivity paint ($\epsilon = 0.95$) to eliminate any thermal differences across the

solid state reference bath by enhancing the longwave radiative absorption and reemmission. Each solid state reference bath was housed in an insulated portable case within a temperature controlled instrument trailer in order to minimize temperature fluctuations in time and across the copper core. The temperature differences within the copper reference sections were less than 0.04K, as observed using multiple PT-100s embedded in the copper cylinders. One solid state reference bath was cooled to approximately 0 °C while the other was heated to approximately 36°C, thereby spanning the range of environmental tem-

peratures observed during LOVE19. Large perturbations away from these target temperatures correspond to periods during which the environmental control for the instrument trailer failed, i.e. after power failures. The reference section temperatures and DTS fiber temperatures within the reference sections are reported in the Zenodo repository for each fiber type.

The DTS devices were operated in a single-ended configuration and thus were calibrated using the single-ended full matrix inversion described in Hausner et al. (2011). All fiber-optic cables were routed through both solid state reference baths both at

the beginning and end of each cable, yielding a total of four reference sections for each fiber (Fig. 4b) at two temperatures. Three reference sections were used for calibration while withholding a reference section for validation and uncertainty analysis. Each reference section was approximately 2 m long, limited by the length of fiber that could be coiled around the copper cylinders. Each reference section was subject to edge effects which were removed, keeping only the segment of fiber that had temporally stable properties and exhibited no temperature gradient. After removing edge effects, the shortest reference sections were 1.5

180    m long consisting of 6 LAF bins for the lower resolution device. For some reference sections, the number of LAF bins is less than the recommended number in Hausner et al. (2011), but the reduction in the number of calibrated points was outweighed by the exclusion of edge effects in the reference sections as evaluated in the validation of reference sections.

The last step in processing the DTS data was to convert from the instrument reported length along fiber (LAF) coordinates to the physical coordinates of the study array (e.g., the conversion from Fig. 4b to 4c). Additionally, the DTS observations were

interpolated to a uniform time step, as the actual time step varied due to instrument idiosyncrasies. The process of mapping to physical coordinates has some uncertainty related to the LAF-step size, which we call $\Delta$LAF, and the instrument's ability to resolve steps in temperature. Temperature artifacts, such as from a fiber holder (Fig. 4b), propagate along the fiber some

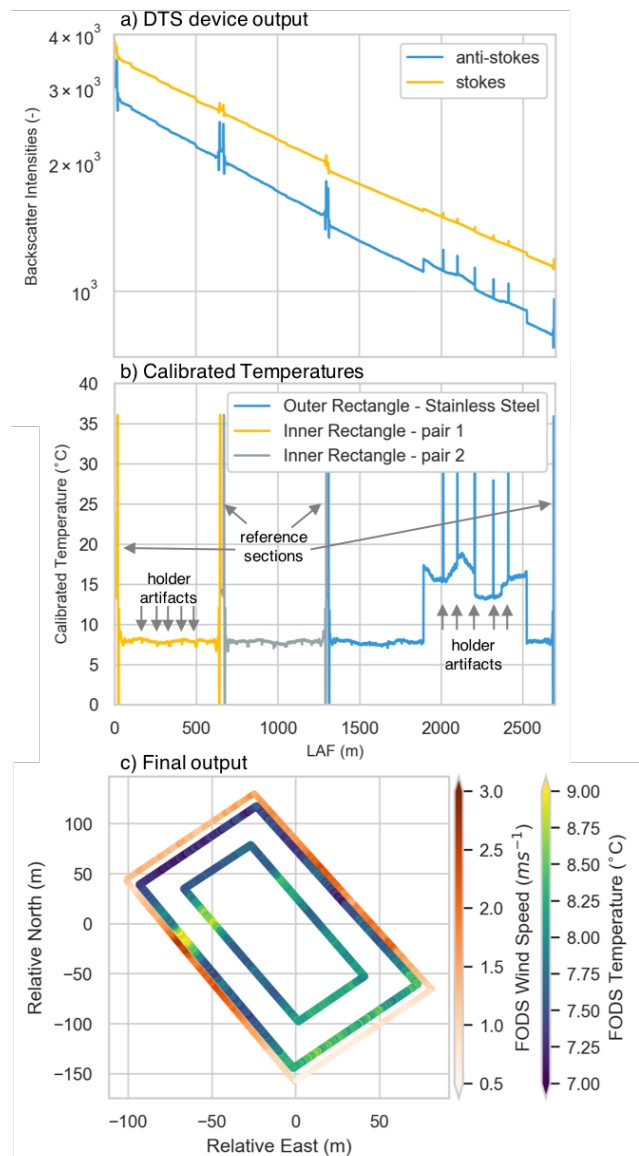

**Figure 4.** An example of the conversion of the DTS data from the (a) instrument reported Stokes and anti-Stokes backscatter intensities. (b) These intensities are calibrated following Hausner et al. (2011) using the reference sections. (a) and (b) share the x-coordinate, which is the instrument reported Length Along Fiber (LAF). The heated sections of the stainless steel fiber can be seen between 1900-2500 m LAF. (c) The calibrated temperatures are mapped from LAF to physically-labeled locations with artifacts from the fiber holders removed and the wind speed perpendicular to the fiber derived from the temperature difference between the heated and unheated sections of fiber (section 5.1). The submeso feature seen in the (c) is analyzed in more detail in section 6.2 and Fig. 11 in order to highlight the utility and novelty of LOVE19.

distance (Pfister et al., 2019). The exact mechanism is unclear but could include conducting heat along the fiber, over-saturation of the optical sensor inside the DTS instrument near sharp temperatures changes (Thomas et al., 2012), and wind artifacts from the cable holder sheltering the cable downstream. These artifacts, in combination with uncertainty in the exact location of a temperature perturbation from the holders, creates an uncertainty in location of at least $\Delta$LAF with distances of 0.5 m being typical. For arrays on the order of kilometers, such as the inner and outer rectangle, we presume this uncertainty is negligible. However, the spatial uncertainty plays a stronger role in shorter sections of fiber, especially when aligning observations for paired fibers as needed for FODS wind speed and direction (section 5).

For FODS wind speed and wind direction (section 5) it is necessary to align sections and interpolate them to a common coordinate. The alignment of the cable-pairs was verified during periods without active heating (Fig. 3). The alignment is particularly critical for vertically-oriented deployments of paired cables, as the vertical gradients in temperature and wind speed make it such that even small a misalignment of size $\Delta$LAF can degrade the derivation of e.g. wind speed by creating a bias between the heated and unheated fibers. Fiber artifacts and alignment have been accounted for in the mapping of all DTS data from LOVE19. The alignment process successfully reduced the bias to 0.001 K for the tower during a 26 hour period without heating (not shown). All data are reported in a site-relative coordinate system. All steps from processing the original instrument output, calibration, mapping to physical coordinates, and aligning heated-unheated and coned sections were performed using software tool pyfocs (version 0.5; Lapo and Freundorfer, 2020).

### 4.1 Horizontal, near-surface FODS arrays

The near-surface FODS array consists of two nested rectangles, inside of which was a 12 m tower with paired heated and unheated fibers, and an 80 m unheated fiber transect running north to south (section 4.2) (Fig. 2a,b,d). The outer rectangle fiber consisted of a 0.84 mm stainless steel cable with a 0.2 mm coating for electrical insulation (FIMT 0.84 SS316L with PE coating with 1 50 $\mu$m multi-mode fiber (OM3), Solifos AG, Fiber Optic Systems, Windisch, Switzerland). The fiber weighed 3 kg km$^{-1}$. The outer rectangle cable was looped such that the same fiber was deployed at two different heights, creating a pair of fibers offset by 0.15 m. The upper stainless steel fiber pair (1.3 m agl) was resistively-heated (Model Heat Pulse System, Silixa, London, United Kingdom), facilitating the derivation of FODS wind speed along the outside of the LOVE19 domain (section 5.1). Due to the directional sensitivity of the FODS wind speed (Sayde et al., 2015; van Ramshorst et al., 2020), the outer rectangle effectively observes the east-west wind speed component along the long side of the rectangle and the north-south wind speed component along the short side. Heating was applied to the outer rectangle by forming a circuit with the heating unit consisting of four parallel sections approximately equal in length. As a result of slight differences in the length of the heated cable across the four sections, the heating rate in W m$^{-1}$ varied around the outer rectangle. The estimated heating rates are included in Lapo et al. (2020a).

The inner rectangle consisted of a twisted pair of PVC cables (Twisted pair - two 900 $\mu$m SBJ with 50 $\mu$m MM fiber (OM3), AFL, Duncan, SC, USA) deployed at 1.3 m agl. The twisted pairs of cable were spliced together with the entire length of cable observed in a single-ended configuration, such that both twisted pairs observed the inner rectangle simultaneously (e.g., Fig. 4). However, in the single-ended configuration, there were unconstrained properties in the calibration, notably the differential

attenuation between the Stokes and anti-Stokes photons. As a result, when comparing the observed temperature between the twisted pairs, there was an LAF-dependent bias ranging between -0.12 K and +0.29 K throughout the inner array. Consequently, Lapo et al. (2020a) only reports the first twisted pair.

The cables for the inner and outer rectangles were spliced together to form one long optical core approximately 2.8 km long (4b). In the direction of the traveling laser light, the PVC fibers of the inner rectangle came first, followed by the the stainless steel fiber. The combined inner and outer rectangle fiber was observed using the XT DTS at a 5 s and 0.254 m resolution, except for between July 16 and July 19 when the high resolution Ultima DTS was employed (Fig 5a,b). During the period with the higher temporal resolution, data were reported as a 1 s average every 2 s (Fig. 3 and 5a,b). Each cable was individually

routed into the solid state reference sections at both the beginning and end of each individual fiber type to enable independent calibrations of each fiber type (4b). As a result of routing the fibers to the reference sections, which were in an instrument trailer  200 m away from the site, in combination with excluding the second twisted pair of the inner rectangle, only 1.62 km of fiber optic cable of the available 2.8 km were retained as being part of the array.

The warm reference section with the largest LAF was withheld for evaluating the DTS calibration. Generally, the calibrated

fiber-optic temperatures have biases slightly higher than published values (Fig. 5a,b). The noticeable deterioration of the validation between July 16 to July 19 can be partially explained by switching to the higher resolution instrument, as this reduced the number of temporal samples due to sampling two channels. However, the increased bias cannot be explained in this way. A possible explanation is that the higher resolution instruments generally perform worse in field conditions than the ruggedized, but lower resolution DTS devices. All FODS array components were sampled by the lower resolution XT DTS on August 1

to reduce maintenance needs. As a result, the inner and outer rectangle temporal resolution became one 5s average every 10 s, since the DTS device can only sample one fiber optic core at a time.

The white noise contribution to the signal variance was calculated using the method from Lenschow et al. (2000). Briefly, in this method, the contribution from white noise is assumed to only impact the autocorrelation of a signal at a lag of zero. One can fit the autocorrelation at larger lags and regress the fit to zero lag. The variance from white noise can be found as the

difference between the observed and modeled autocorrelation at zero lag. Using this method the noise variance was estimated for each fiber type during a period without heating on the calibrated data. This information regarding the noise contribution to the DTS signal at a given LAF (Fig. 6), in combination with the evaluation of the DTS calibration (Fig 5), enables users of the data to filter for periods with sufficiently large signal-to-noise ratios for detecting quantities of interest, as in Peltola et al. (2021). As a result of the ordering of fibers and observing the inner and outer rectangle as one single, long optical core,

the outer rectangle observations had a larger uncertainty than those of the inner rectangle (Fig 6a). The validation reference sections always had a higher standard deviation in time, $\sigma(T)$, than the reference sections used for calibration.

## 4.2   FODS-cross

The FODS-cross, NS-transect, and 12 m tower (Fig. 2a,b,d,e), were observed using the high-resolution DTS instrument with a 1 s and 0.127 m sampling resolution. The NS-transect was only an unheated fiber while the 12 m tower and FODS-cross were

composed of unheated and heated pairs of fiber. The optical cable that was loosely buffered inside a high-resistance stainless

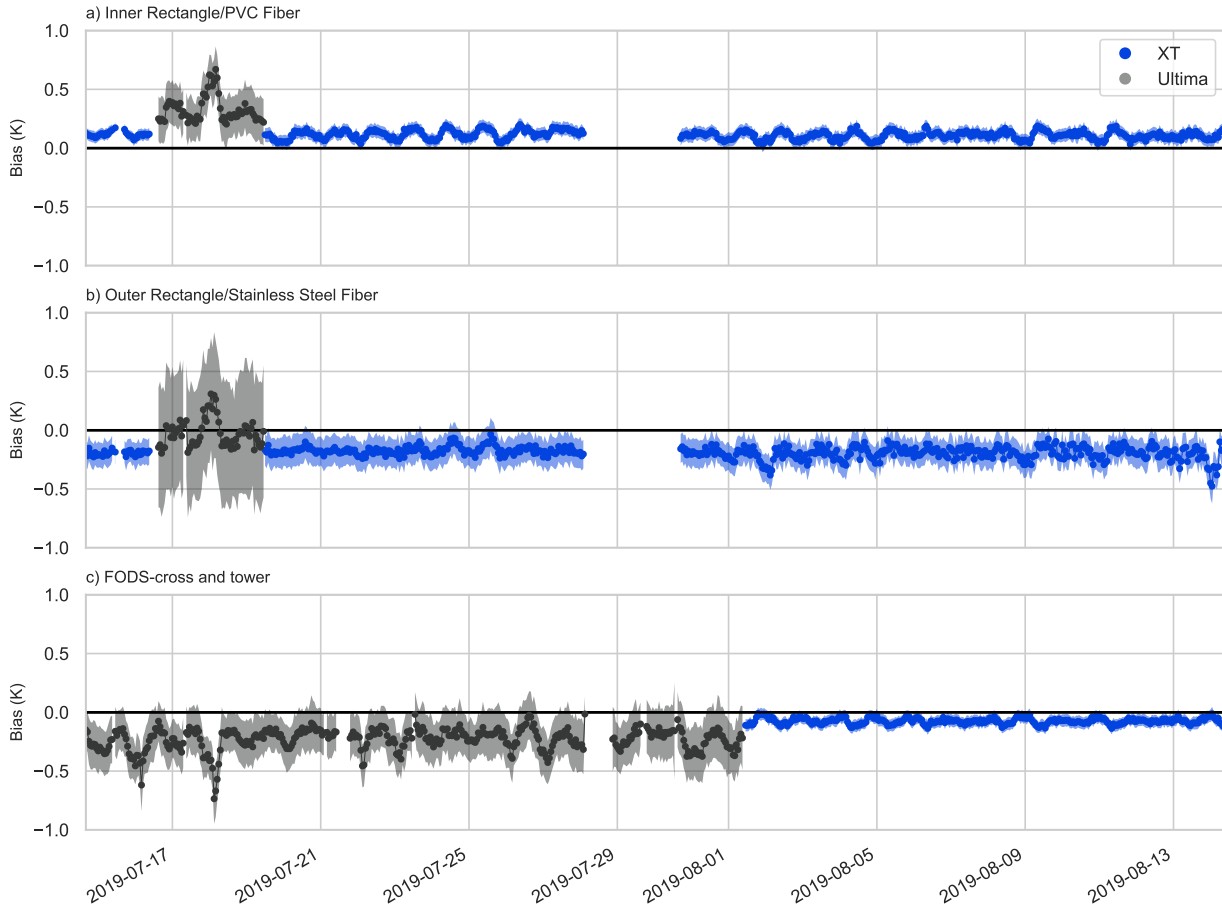

**Figure 5.** The bias in calibrated DTS temperatures relative to the reference PT-100 in the validation reference section. Biases are averaged over all spatial samples in the reference section and aggregated from the instrument time step to 1 hour. Solid lines are the mean bias and the shaded regions are the standard deviation of the bias within the hour. Bias time series are shown for (a) the inner rectangle, (b) the outer rectangle (both described in section 4.1), and (c) the FODS-cross (described in section 4.2). The black and blue colors are provided as guidance for selecting lower and higher quality data, respectively.

steel sheath filled with gel (outer diameter 1.32 mm, Model C-Tube (OM3), Brugg, Switzerland). The fiber-optic cable was coated by a 0.2 mm thick polyethylene (PE) jacket for electric insulation. The fiber weighed 5 kg km$^{-1}$. The FODS-cross consisted of an identical fiber type with small, directional microstructures attached by injection molding (Lapo et al., 2020b). The coned fiber was spliced to the unconed fiber to form a single optical core. Both fibers contained four optical cores, of which only the one with the shortest LAFs was used due to increasing instrument noise with longer LAF (Fig 6).

A single optical core was 895 m. However, the fiber was routed between the study site and the reference sections (approximately 200 m). This routing, in combination with the length of unused coned fiber and spare fiber kept at critical points in the

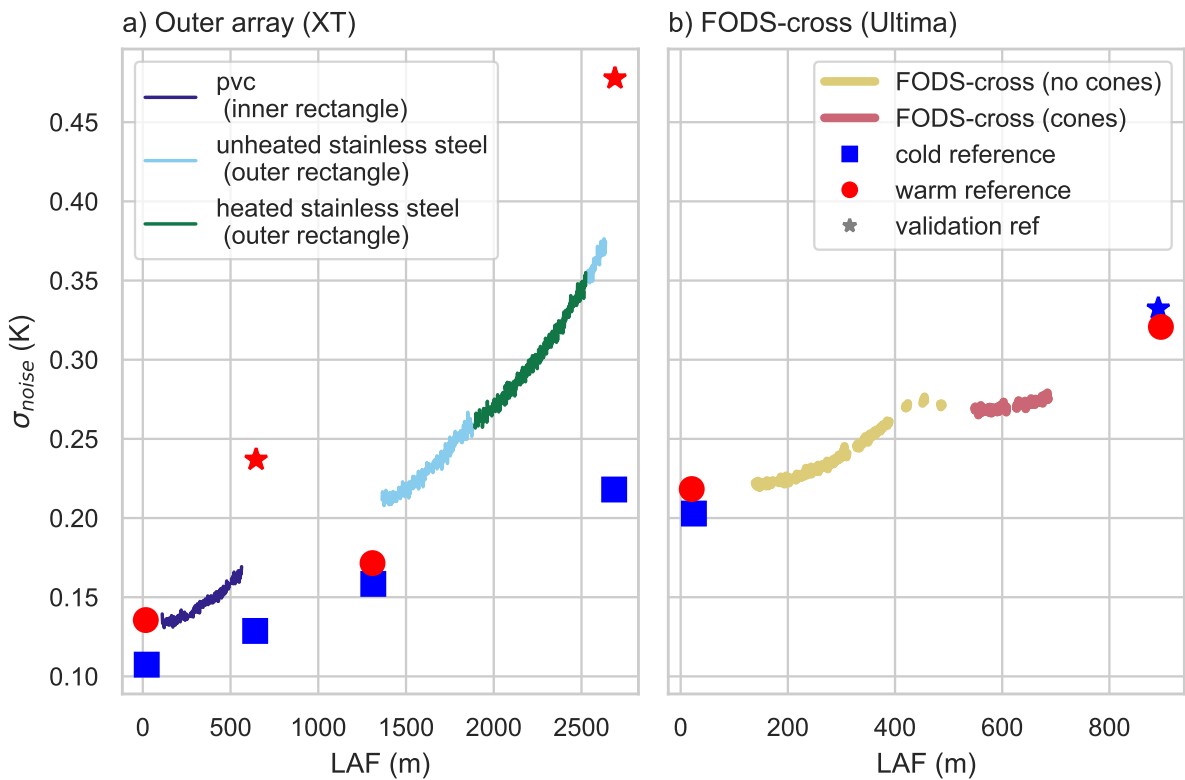

**Figure 6.** Noise contribution to the signal, $\sigma_{noise}$, using the method from Lenschow et al. (2000) for (a) the XT observing the inner and outer rectangle and (b) the Ultima observing the tower and FODS-cross for a 26 h period without heating. For the Ultima, $\sigma_{noise}$ was adjusted by $\sqrt{10}$ to account for differences in the temporal (5 s vs 1 s) and spatial resolution (0.254 m vs 0.127 m) between the XT and Ultima, enabling comparisons. The standard deviation of the reference sections in time is shown for each fiber with the validation reference section highlighted using a star symbol. The Lenschow noise estimate and the standard deviation of the reference sections temperatures in time agreed, so only the standard deviations of the reference sections are provided for clarity.

array in case of a break, resulted in only observing 290 m of fiber optic core as part of the actual array out of the available 895 m.

These FODS elements were observed in a single-ended configuration and calibrated as a single length of fiber. The cold reference section with the largest LAF was withheld for validation. The calibrated temperatures for this component generally have more variability and slightly larger biases than the inner and outer rectangle (Fig. 5). Additionally, the Ultima observations had a larger noise variance than the XT for similar LAF values (Fig. 6). Users of the data should account for this enhanced noise when employing these data.

The tower fibers were deployed 0.5 m away from the tower and oriented to the northeast. The fibers were attached at the top and bottom of the tower by gently looping the fiber multiple times around a plastic disk with a 0.15 m diameter, which

gives the distance between the heated and unheated fibers, in order avoid bend artifacts. The tower fibers were gently pulled to be taught, but not too tight as to create a bend or strain artifact. Consequently, the fibers could sway under sufficiently strong winds, but this was not found to create an artifact. There do appear to be some artifacts on the tower fiber that vary with the time of day, e.g. as found in section 5.1.

Within the FODS-cross (Fig. 2d,e), quartets of fibers consisting of a pair of fibers with small cones oriented in opposing direction, for observing wind direction, and paired heated and unheated unconed fibers, for observing wind speed and air temperature, were strung within a rectangular frame. The paired fibers were horizontally separated by 15 cm (Fig. 2e). Orthogonal quartets of fiber were deployed at heights of approximately of 0.5 m (within the grass canopy), 1 m, and 2 m. Each section was vertically offset by 0.25 m from its orthogonal counter-part. The exact coordinates are included within Lapo et al. (2020a).

The FODS-cross was intended specifically as a test-bed and the first environmental deployment of the FODS wind direction method. Preliminary lab work was able to successfully observe distributed wind direction in the one-dimensional flow of a wind tunnel (Lapo et al., 2020b). Based on the result from this preliminary work, PE cones were attached to the stainless steel fiber using injection molding. Cones were 12 mm in diameter and height with a 2 cm separation, consistent with the optimal cone construction determined by Lapo et al. (2020b). The initial results from the FODS wind direction method are the subject of on-going work and will be published elsewhere.

A common problem with DTS is determining the resolvable scales of the technique in space and time simultaneously (e.g., Thomas et al., 2012). We address this concern by describing the resolvable scales for the unheated tower fiber using the spectral coherence, $C$.

$$C = \frac{|F_{xy}^2|}{F_{xx}F_{yy}} \tag{1}$$

where $F_{xy}$ is the cross-spectral density estimate between the quantities $x$ and $y$, and $F_{xx}$ and $CF_{yy}$ are the power spectral densities of these quantities. $C$ is analogous to the linear correlation between two spectra at a given frequency (Stull, 1988). It varies between values of 1 and 0, with higher values when the phase and amplitude of the spectra are consistent through time and decreases when these relationships are inconsistent. Using this property, $C$ is used to characterize how coherent the DTS and sonic anemometer spectra are at a given time scale. We conservatively estimate the finest resolvable time scale of the DTS as the the first time scale at which $C$ drops under a value of 0.01, meaning that the two spectra are spectrally decorrelated at finer time scales. This process is repeated using increasingly large spatial averages of the DTS, which decrease the noise of the DTS signal and improves $C$. Thereby, this method simultaneously characterizes the resolvable time scales for a given amount of spatial averaging.

This process is performed for the unheated tower fibers observed by the Ultima between 20 and 27 July and compared to the CSATs at heights of 1.25 m and 4 m (Fig. 7 as the 0.5 m and 12 m sonic anemometers were slightly outside the vertical range of the tower fibers.

At the finest spatial resolution of the Ultima, the FODS observations were not spectrally coherent at time scales smaller than 14-15 s, which forms a minimally resolvable time scale. As the amount of spatial averaging increases, the minimum resolvable

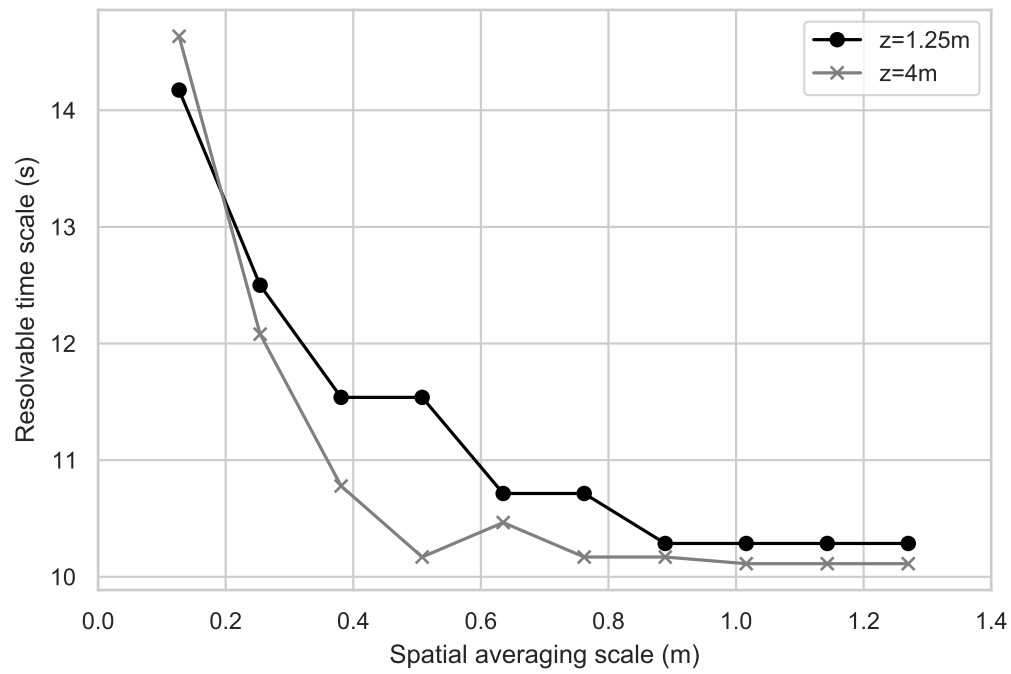

**Figure 7.** The scale at which the spectral coherence (eq. 1) between the sonic anemometer and unheated tower temperatures falls below 0.01 for increasing spatial averaging centered on the LAF bin nearest the indicated sonic anemometer.

time scale drops to a value between 10-11 s for spatial averaging of 0.5 m (4 $\Delta$ LAF) for the 4 m comparison and 0.89 m (7 $\Delta$ LAF) for the 1.25 m comparison. Any additional spatial averaging does not improve the minimum resolvable time scale. Thus, one can conclude that the unheated tower fibers have a conservative minimum resolvable time scale of 10-11 s and spatial scale of 0.5-0.8 m.

     This minimum resolvable time scale is longer than in previous studies (e.g. Thomas et al., 2012; Peltola et al., 2021, >6.4
310   s and 2-3 s respectively). This difference may be due to differences in the type of fiber, calibration, or method of determining scales. The coherence method quantifies the variance shared by the reference and DTS spectra, while the other two methods are based on similar spectral shapes. For this reason, we describe the coherence method as a conservative estimate. The coherence method unfortunately cannot be completed for any other locations in the FODS array, as it is sensitive to the time lag between points, which can vary for physical reasons between distant points, e.g. between the CSATs and the inner rectangle. This
technique serves as the first simultaneous estimate of resolvable spatial and temporal scales of the DTS technique.

## 4.3 FlyFOX-V

The Flying Fiber Optic eXperiments - Voitsumra (FlyFOX-V) were performed using a vertical deployment of the fiber attached to a tethered balloon. These flights enabled observing air temperature between 2 m and 200 m agl (Fig. 2c) at a fine resolution (3 s-10 s and 0.127 m-0.254 m), yielding an unprecedented level of observational detail of the structure of the wwSBL. While the tethered FODS technique has been used previously for atmospheric profiling (e.g. Keller et al., 2011; Higgins et al., 2018), these previous studies were fundamentally limited in duration (30 min and 17 h), spatial (0.25 m and 1.0 m), and/or temporal resolution (20 s and 5 min). FlyFOX-V removed these limitations through its fine resolution and longer flight times, successfully resolving the fine-scale structure of the morning transition for the first time (Fritz et al., 2021). We anticipate these data and overall technique will enable similar breakthroughs by resolving processes that are otherwise missing in the current repertoire of atmospheric profiling techniques.

The tethered FODS experiments were conducted with a twisted-pair of PVC fibers (Twisted pair - two 900 um SBJ with 50 um MM fiber (OM3), AFL, Duncan SC), observed in a single-ended configuration. The fibers were run through two reference water baths, one warm and one cold, both at the beginning and end of the length of fiber tethered to a balloon, allowing calibration following the same procedure as for the other FODS components. The reference water baths were monitored with waterproof temperature sensors (RBRsolo[3] T, RBR, Ottawa, ON, Canada). One bath was heated using an aquarium heater while the other was an ambient temperature bath. Both baths were continually mixed using aquarium pumps.

The twisted pair of fibers were spliced together at the turn around point at the top of the airborne section of the tethered profile. The airborne profile was at the end of the first twisted pair, called the ascending pair at an LAF between approximately 900 m and 1100 m, and the beginning of the second twisted pair, called the descending pair, at an LAF of approximately 1100 to 1300m. The exact LAFs depended on the flying altitude of the balloon. Calibrated temperatures had a bias of ≈0.1 K, which was higher than expected (Hausner et al., 2011). While this bias was acceptable, the pair of fibers were found to have clearly different differential attenuation values. Consequently, there was an LAF-dependent disagreement between the ascending and descending cables as large as 0.3K near the surface. The single-ended calibration inhibited accounting for the change in differential attenuation between the pairs of fibers, leading to these disagreements. The original intent of this setup was to provide duplicate observations at every point. In hindsight, the decisions to splice pairs together at the top of the profile, sampling the ascending cable after 1000 m of LAF thereby increasing the noise of the signal, and using a single-ended set-up were design flaws that degraded the signal quality. These results suggest that even higher quality observations with the FlyFOX approach are possible. Only the ascending fiber is provided in the data repository.

A custom-designed tethersonde observing air temperature, relative humidity, and air pressure (Model BME280, Bosch sensortec GmbH, Reutlingen, Germany) in addition to wind speed using a hot-wire anemometer (Model Rev C, Modern Device, Providence, USA) contained in a 3D-printed housing was deployed immediately below the tethered balloon, at the top of the FODS profile. Biases in the tethersonde pressure were removed by comparing to high-quality observations (Digiquartz Nano-Resolution Barometers Model 6000-16B, Paroscientific, Redmond, WA, USA) when the balloon was at the surface.

**Table 1.** Details for the FLYing Fiber Optic eXperiment - Voitsumra (FlyFOX-V).

| Flight | times | DTS device | Notes |
|--------|-------|------------|-------|
| 18 July | 3:08Z-6:54Z | Ultima | Fog during the first several hours of flight. Winds less than $2ms^{-1}$ within lowest 150m. |
| 22 July | 3:18Z-6:10Z | XT | Relative humidity >90%, no fog, winds less than $2ms^{-1}$ within lowest 150m. Wind direction within SBL southerly. Cloud cover between $\frac{5}{8}$ to $\frac{7}{8}$. |
| 23 July | 2:59Z-6:30Z | XT | Relative humidity >90%, no fog, winds less than $2ms^{-1}$ within lowest 150m. Wind direction within SBL southerly. |
| 24 July | 18:13Z-19:55Z | XT | Qualitatively higher wind speeds but SODAR observations were not available to verify, lower and more variable balloon height. |
| 26 July | 3:21Z-6:24Z | XT | Relative humidity >90%, no fog, winds less than $2ms^{-1}$ within lowest 150m. Wind direction within SBL northerly. |

FlyFOX-V observed four morning transitions and one evening transition (Table 1). All flights except 18 July were observed with the lower-resolution XT DTS device. As the DTS device observed both the inner/outer rectangle as well as FlyFOX-V, the temporal resolution for both components was a 5 s temporal average every 10 s during the flights. The flight on 18 July was observed with a second high-resolution Ultima DTS device simultaneously with the outer/inner rectangle, yielding a temporal resolution of 1 s averages every 3 s and a 0.127 m spatial sampling resolution. All flights occurred on mornings with relatively little cloud cover and no fog except for 18 July (Table 1). This flight was characterized by low-lying fog (which can be seen in Fig. 2c). The sun rose locally on the launch area at 4:50 due to the effect of local shading from nearby topography and trees, which considerably impacts the dynamics of the morning transition (Fritz et al., 2021).

The DTS temperatures along the tethered balloon profile were converted from an LAF coordinate to a height coordinate, $z$, and from dry-bulb temperature to virtual potential temperature, $\theta_v$. An iterative solution to the hypsometric equation was employed using observations of pressure and relative humidity at the surface and the tethersonde as well as the DTS observations along tethered profile (see supplemental material in Fritz et al., 2021).

$$z = \frac{R\overline{T_v}}{g} ln \left( \frac{p_{sfc}}{p(z)} \right) \tag{2}$$

where $z$ is the height above the surface, $R$ is the specific gas constant for dry air, $\overline{T_v}$ is the mean virtual temperature between the surface and height $z$, $p_{sfc}$ is the pressure at the surface, and $p_z$ is the pressure at $z$. The height coordinate was linearly interpolated between the surface and the top of the profile.

## 5  Fiber Optic Distributed Sensing of Wind Speed and Direction

Both the FODS wind speed and direction methods are based on the temperature difference between pairs of fibers. The data available in Lapo et al. (2020a) are only the calibrated temperatures. In both cases the decision to report calibrated temperatures instead of derived quantities enables users of the LOVE19 data to refine their own derivation of the quantity in question. For wind speed, refinements are possible but the wind speeds presented here can be trivially recovered using pyfocs (Lapo and Freundorfer, 2020) following the example scripts provided in the data repository (Lapo et al., 2020a), whereas for FODS wind direction the method is still experimental and is the subject of on-going work. For both methods the heating rate is provided.

### 5.1  Wind Speed

Distributed wind speed is observed using a pair of cables, with one of the cables resistively heated and the other cable unheated, effectively creating a distributed hot-wire anemometer (Sayde et al., 2015; van Ramshorst et al., 2020, S15 and vR20 respectively). The difference in temperature between the two fibers is a function of the wind speed orthogonal to the fiber. Larger temperature differences indicate a slower wind speed and smaller temperature differences indicate a faster wind speed. As the wind speed increases, the temperature difference can become small enough that DTS instrument uncertainty contaminates the signal. With sufficiently strong winds the temperature difference can disappear entirely, leading to a saturation effect. This saturation effect was found to be one of the primary sources of uncertainty in previous work (Sayde et al., 2015).

There are two versions of the FODS wind speed derivation. The original version from S15 is

$$U_N = \frac{0.5P\pi^{-1}r^{-1} + (S_b + S_d + \rho S_t) + \epsilon L_{in} - \epsilon\sigma T_s^4 + \frac{1}{2}c_p\rho\frac{dT}{dt}}{-C(2r)^{(m-1)}Pr^n\frac{Pr}{Pr_s}^{1/4}K_a\nu^{-m}(T_s - T_f)} \tag{3}$$

where $U_N$ is the wind speed orthogonal to the fiber pair, $P$ is the heating rate in Wm$^{-1}$, $r$ is the fiber's outer radius, $S_d$, $S_d$, and $S_t$ are the direct, diffuse, and surface reflected shortwave respectively, $L_{in}$ is the mean of the downwelling and upwelling longwave irradiances, $\epsilon$ is the fiber's emissivity, $T_s$ is the heated fiber's temperature, $c_p$ is the specific heat capacity of the fiber, $\rho$ is the fiber's density, $C$, $m$, and $n$ are flow dependent coefficients, $Pr$ and $Pr_s$ are the Prandtl numbers for air temperature and the heated fiber respectively, $K_a$ is the thermal conductivity of air, $\nu$ is the kinematic viscosity of air, and $T_f$ is the temperature of unheated fiber (i.e., air temperature). In LOVE19 we remove some of the complicating factors in FODS wind speed by using identical fibers, as in vR20, such that the shortwave irradiances drop out and the equation simplifies to

$$U_N = \frac{0.5P\pi^{-1}r^{-1} + \epsilon L_{in} - \epsilon\sigma T_s^4 + \frac{1}{2}c_p\rho\frac{dT}{dt}}{-C(2r)^{(m-1)}Pr^n\frac{Pr}{Pr_s}^{1/4}K_a\nu^{-m}(T_s - T_f)}. \tag{4}$$

vR20 additionally suggest improvements to the representation of the convective heat transfer, yielding

$$U_N = \frac{0.5P\pi^{-1}r^{-1} + \epsilon L_{in} - \epsilon\sigma T_s^4 + \frac{1}{2}c_p\rho\frac{dT}{dt}}{-C(2r)^{(m-1)}Pr^n K_a\nu^{-m}(T_s - T_f)} \tag{5}$$

with the values for $C$, $m$, and $n$ adjusted relative to S15. Both versions of the wind speed expression are available in pyfocs (Lapo and Freundorfer, 2020). All FODS wind speeds shown in this manuscript were derived following vR20.

As the vR20 and the vertically-oriented fibers have not been tested in an environmental application, the FODS wind speed along the 12 m tower were evaluated. In previous studies of FODS wind speed, great care is taken to adjust the evaluation of FODS wind speed for the angle of attack (Sayde et al., 2015; Pfister et al., 2019; van Ramshorst et al., 2020). For the FODS wind speed along the 12 m tower, this complicating factor is not relevant. For flows over flat surfaces, the near-surface flow deviates only little from horizontal due to the small time-averaged vertical wind speed. Thus, over flat surfaces like in LOVE19 the vertical orientation of the cables removes the angular dependence for deriving the horizontal wind speed since the angle of attack is orthogonal with respect to the fiber. However, in hot wire anemometery, neglecting the turbulent perturbations parallel to the hot wire can lead to erroneous results even when the magnitude of that wind component is small (**?**), which is an effect not considered in previous studies of FODS wind speed.

Initial analysis of FODS wind speed revealed time varying biases. In previous work with FODS wind speed, biases were attributed to angular effects. As angular effects could be neglected, it was suspected that instead there may be unresolved energy balance factors in the wind speed derivation (eq. 4). Regardless of the exact source of error, the heating rate can be treated as a tuning parameter that can be adjusted in order to compensate for any errors and thereby optimize FODS wind speed.

To achieve this, the heating rate, $P$, was varied over a range of values while solving eq. 4 for U for the period with heating, July 15 to July 28 (Fig. 3). The derived wind speeds were then compared to the tower sonic anemometry observations. The LAF bin nearest to the 1.25m sonic anemometer was evaluated to yield a mean bias and to find the linear slope between the two observations (Fig. 8a-d). The evaluation was stratified according to day (0700-1600) vs night (2000-0600) and cloudiness regimes. Cloudiness regimes were defined using the ceilometer-derived cloud cover octets ($\frac{0}{8}$-$\frac{1}{8}$ were clear, $\frac{2}{8}$-$\frac{6}{8}$ were partially cloudy, and $\frac{7}{8}$-$\frac{8}{8}$ were cloudy). All nighttime evaluations yielded the same optimal heating rate of 4.5 Wm$^{-1}$ for FODS wind speed regardless of sky condition, only slightly higher than the observed heating rate of 4.3 Wm$^{-1}$ (Fig. 8b). In contrast, the daytime evaluations indicate that sky condition affected the derivation of FODS wind speed, as seen by cloud cover-dependent optimal heating rates (Fig. 8a).

FODS wind speed during the daytime with cloudy conditions tended to have more scatter and overestimated the wind speed relative to FODS wind speed during the day with clear conditions for the same heating rate (Fig. 8e,f). In S15, FODS wind speed tended to underestimate wind speed during stronger winds as a result of the saturation effect when the heating rate was not strong enough to maintain a sufficiently large temperature difference between the heated and unheated fibers. Our observations did not confirm this finding, as the FODS wind speed overestimates stronger winds during cloudy conditions. A heating rate of 4.5 Wm$^{-1}$ is two times larger than the heating rate used in S15. The difference in temperature between the heated and unheated fibers did not fall below 4.2K, with a mean value of 8K and a maximum of 31K, suggesting that the saturation effect can be solved with sufficient heating of the fiber.

These temperature differences are larger than those reported in either S15 or vR20. vR20 and S15 suggested that the largest temperature differences would lead to forced convection and thus an error in the wind speed calculation as this term is not con-

sidered. This error should be prevalent at the lowest wind speeds when the temperature differences were the largest. However, FODS wind speed evaluates quite well at low wind speeds suggesting this term is less relevant for this atmospheric deployment.

The decreased performance of FODS wind speed during the day and clear-sky conditions indicates the existence of radiation artifacts. The unheated fiber shows a clear shortwave heating artifact during the day (not shown, Sigmund et al., 2017). As a result of this heating, the unheated fiber may not be at air temperature. The unheated fiber potentially experienced its own convective heat loss, giving $T_f$ in equation 4 a small error that depended on wind speed. Additionally, radiative and flow obstacles can be seen when examining the entire FODS wind speed profile along the tower (Fig. 8i-l). During the day with clear skies, a deviation from the expected logarithmic wind speed profile occurs near 8 m agl, while at night an artifact appears around 4 m. As a result of these artifacts, selecting an optimal heating rate according to the evaluation at 1.25 m creates biases up to 0.25 ms$^{-1}$ when evaluating against the 4 m sonic anemometer for the same heating rate. We suggest that the 12 m tower FODS wind speed cannot resolve vertical gradients with a change in wind speed smaller than this value.

Liquid water intercepted by the FODS cables during significant rain events presents a temporary source of uncertainty for FODS wind speed as it affects the term $(T_s - T_f)$ until all water attached to both unheated and heated cables has evaporated. Heated cables dried quicker compared to unheated cables, and drying time was shorter during day than night. The exact drying time was not further evaluated, but was on the order of tens of minutes, during which users are advised to exercise caution when investigating FODS wind speed. High relative humidity without intercepted liquid water present on the FODS cables was not found to have an effect on FODS performance.

## 5.2 FODS wind direction

The FODS-cross used orthogonal segments of coned and heated-unheated fiber pairs for the goal of fully resolving two dimensional atmospheric flow on a distributed basis. The FODS wind direction method is predicated on a similar argument as FODS wind direction. A pair of resistively-heated fibers with microstructure cones attached in opposing directions induce a convective heat loss that is sensitive to the wind direction along the fiber (Lapo et al., 2020b). As a result, the temperature difference between the coned-fiber pairs is related to wind direction. However, this approach has only been demonstrated in wind tunnel tests for one-dimensional flows along the fiber. We sought to refine the FODS wind direction technique for two-dimensional flows by taking advantage of the coned fibers with orthogonal orientation. LOVE19 was an environmental test-bed for this approach and is the subject of ongoing work to be published soon.

## 6  Examples of observed submeso-scale structures

These data are intended to be used to study submeso-turbulence interactions in the wwSBL. We demonstrate two use cases for the LOVE19 data, specifically highlighting how the novel FODS data observed submeso-scale features which cannot be observed using traditional sensor networks. These examples are provided to help elucidate the utility of spatially-distributed observations, but are not exhaustive in their analysis, as that is beyond the scope of this data paper. In the first example, a column of vertical observations are presented to highlight how FlyFOX-V can fill in details missing from the ground-based

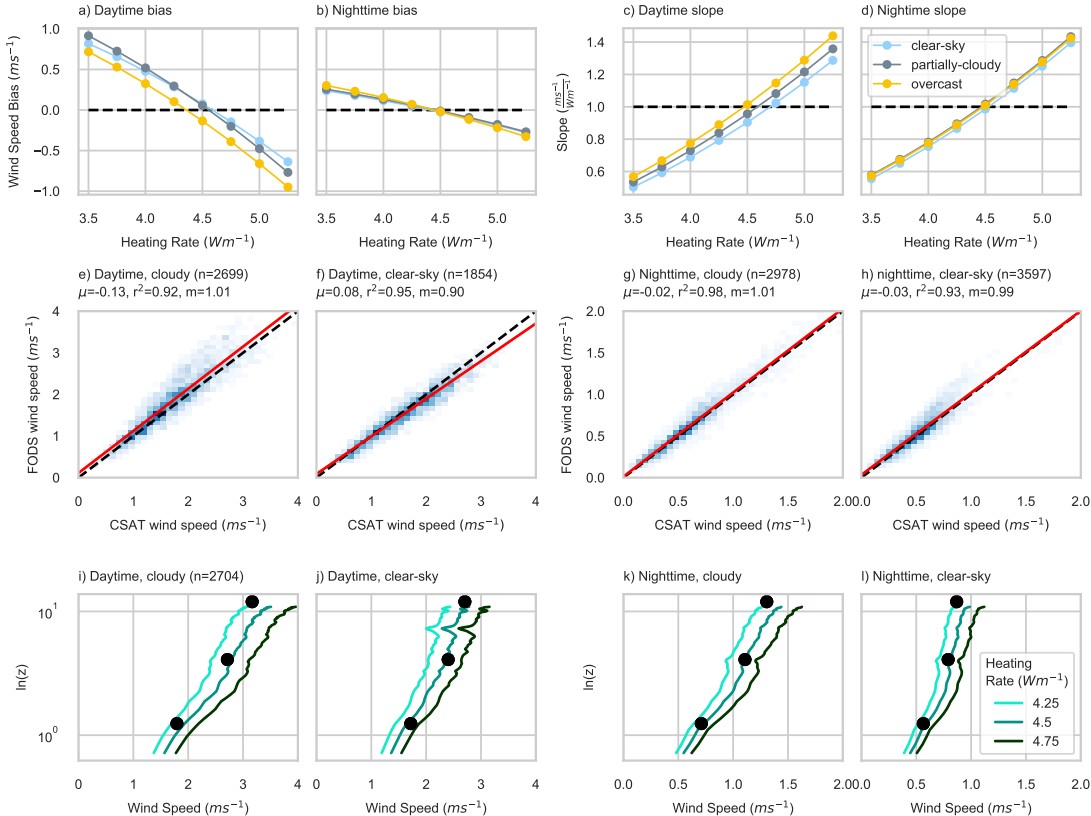

**Figure 8.** The heating rate of the fiber-optic cable was varied when deriving the FODS wind speed following van Ramshorst et al. (2020). The (a,b) bias and (c,d) sloped in FODS wind speed evaluated against the horizontal wind speed observed by the 1.25 m sonic anemometer. Data were stratified according to solar light and cloud regime. For the heating rate of 4 $Wm^{-1}$, (e-f) two-dimensional histograms of the evaluation are shown. The caption indicates bias ($\mu$), correlation coefficient ($r^2$), linear slope ($m$), and the number of minutes evaluated for each regime classification. For these same regime classifications, (i-l) vertical profiles of wind speed are shown for a varying heating rate. The dots indicate the mean horizontal wind speed observed by the sonic anemometers at 1.25 m, 4 m, and 11.99 m.

remote sensing (Fig. 9). Near the surface, FODS observations along the tower and surface array demonstrate the unique ability to track these wave motions as they propagate horizontally (Fig. 10). In the second example, a near-surface submeso-scale structure is shown which propagates across the study area (Fig. 11). The submeso-scale structure co-occurs with meandering of the horizontal wind direction and an intermittent burst of turbulence near the surface.

## 6.1 Internal Gravity Waves

During the 18 July flight a layer of fog was present near the surface initially, but lifted to approximately 20 m height by 4:15 (the low lying fog can be seen lifting above the surface in Fig. 2c). The top of the SBL can be characterized using the change

in temperature gradient from stable to neutral in the FlyFOX-V data (Fritz et al., 2021). During this period the top of the SBL
was significantly non-stationary due to the waves, but was typically around 80 m (Fig. 9e). The SODAR-RASS observations
indicated directional shear across the top of the SBL, from southwesterly (up-valley) in the SBL to northeasterly (down-valley)
in the residual layer (Fig. 9b) and wind speed shear shear within the residual layer ( 150-250 m; Fig. 9c).

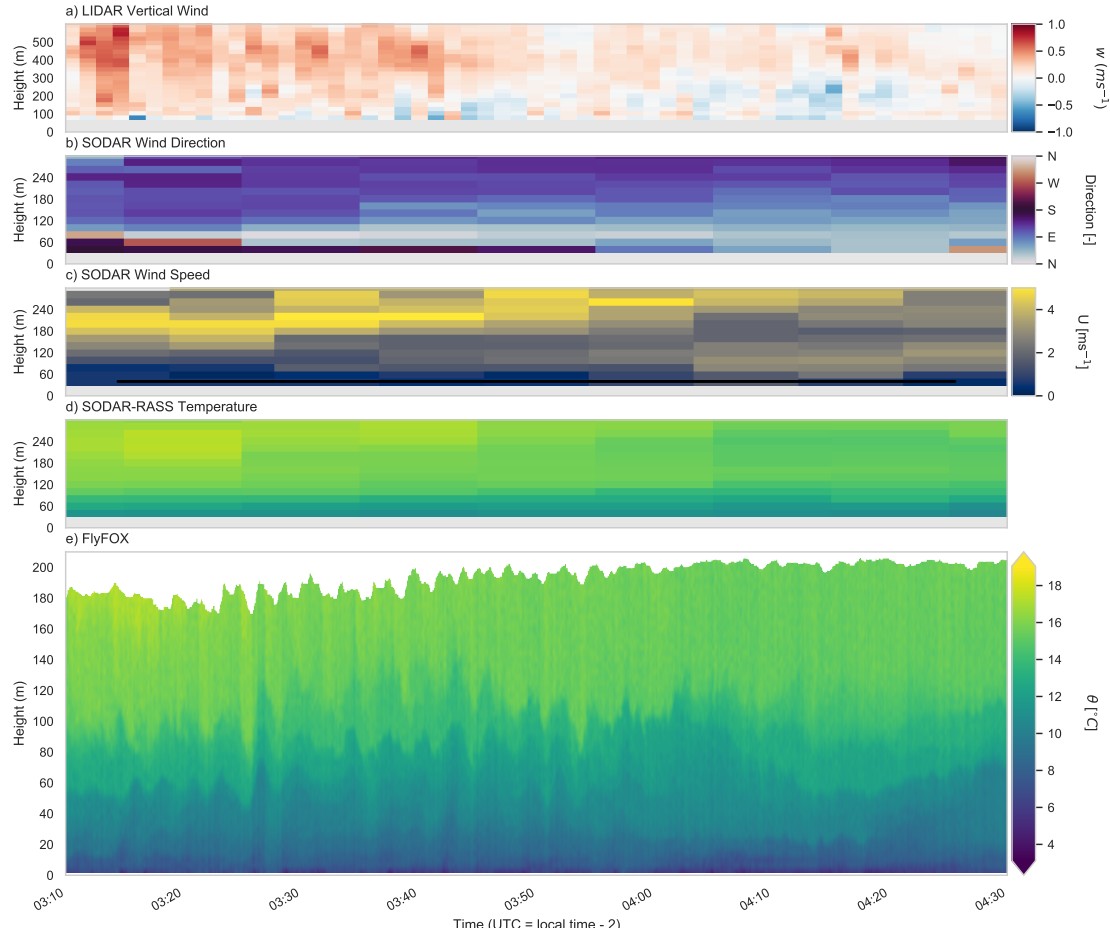

**Figure 9.** The LOVE19 "column" during the first FlyFOX-V on July 18 2019. (a) The LIDAR vertical stares yielded vertical wind speeds,
which were aggregated to 84s. The SODAR derived (b) horizontal wind direction, (c) horizontal wind speed, and (d) the SODAR-RASS
potential temperature during the flight. (e) The FlyFOX-V data, converted to potential temperature and height above the surface.

    Between 3:20 to 3:50 the wave action was vigorous and easily identified visually. Examining the FlyFOX-V observations
reveals that the waves spanned the entire depth of the SBL (Fig. 9e) and potentially even into the residual layer where they
were detected in the LIDAR data as vertical wind speeds oscillations (Fig. 9a). As the stronger flow above the top of the SBL
lifted after 4:00 (Fig. 9c) the wave action diminished. FlyFOX-V observed that the top SBL decreased during this period (Fig.
9e), a feature that the SODAR-RASS largely missed due to the coarser resolution.

These waves can be observed using the surface DTS observations (Fig 10). They are clearly evident in the surface DTS array as alternating longer duration warm perturbations interspersed with shorter duration cold perturbations (Fig. 10a,b). These wave-like oscillations are most clearly organized between 3:20 and 3:45, during which time the cold perturbations were associated with the increase in FODS wind speed along the outer rectangle (Fig 10c). The vertical structure of these waves was observed by the tower FODS observations of temperature and wind speed, with which these waves can be seen coupling to the surface. The distinct cold-air perturbation appears to be cold air lifted from near the surface (Fig. 10d). Similar to the FODS wind speed along the outer rectangle, the cold-air perturbations were associated with bursts of horizontal wind speed that spanned the height of the tower (Fig. 10e). After 3:50, the wave activity became less organized and other submeso-scale modes appear to have emerged. Coincident with this, the tower FODS temperature observed the reemergence of a decoupled near-surface layer while tower FODS wind speed observed calm conditions.

A preliminary analysis with the observed momentum flux calculated at a 1 min perturbation time scale (Fig. 10f) indicates that some of these waves transported relatively large amounts of momentum. The vertical wind speed observed at 20 Hz (Fig. 10g) suggests that the wave activity was fairly well-organized at 12 m, where clear vertical oscillations were observed. Nearer the surface, these oscillations became distorted. Gravity waves have been found to be distorted by near-surface stability, which can lead to turbulent transport of mass and energy (Sun et al., 2015; Cava et al., 2019).

Resolving the wave-turbulence interactions is a major goal of stable boundary layer research (Sun et al., 2015) and the FODS data presented here are a promising avenue for investigating these phenomenon. Traditional observations are unable to resolve the vertical interactions highlighted in this simple demonstration (e.g., compare Fig. 9a,d to Fig. 9e), with the surface DTS arrays providing the unique ability to observe the horizontal propagation of these waves. Even other cutting-edge methods like drone observations would miss the wave events presented here due to the relatively long repeat time between flights (e.g., Pillar-little et al., 2020; Kral et al., 2020) and the LIDAR could only resolve the wave action when it was most vigorous and sufficiently above the surface. FODS techniques, such as FlyFOX and DTS arrays, can connect the surface layer to the upper boundary layer at an unprecedented spatial and temporal resolution, filling a critical need in boundary layer research.

## 6.2 Near-surface submeso motions

The second example focuses on an intermittent burst of turbulence associated with a near-surface, spatially-discrete submeso-scale structure (Fig. 11). A movie of this event can be found as a video supplement to this manuscript. While the exact structure is difficult to define precisely through a visual inspection, it is clear that the perturbation is a transient temperature structure with a leading cold perturbation and trailing warm perturbation (Fig. 11a,b,d,f) between 60-80 m wide. Associated with the temperature structure was a maximum in horizontal wind speed (Fig. 11d,e,g,h,i,l,m). This wind speed maximum was located between 1 m and 4 m above the surface (Fig. 11l,m). The temperature component of the structure extended up to approximately 6 m agl, although this determination depends on which features are considered part of the structure as opposed to the background state.

The structure traversed the site starting in the northwest corner traveling towards the southeastern corner (see video supplement, Fig 10g-i). The structure propagated by the tower in approximately 2 minutes, suggesting a mean advective velocity of

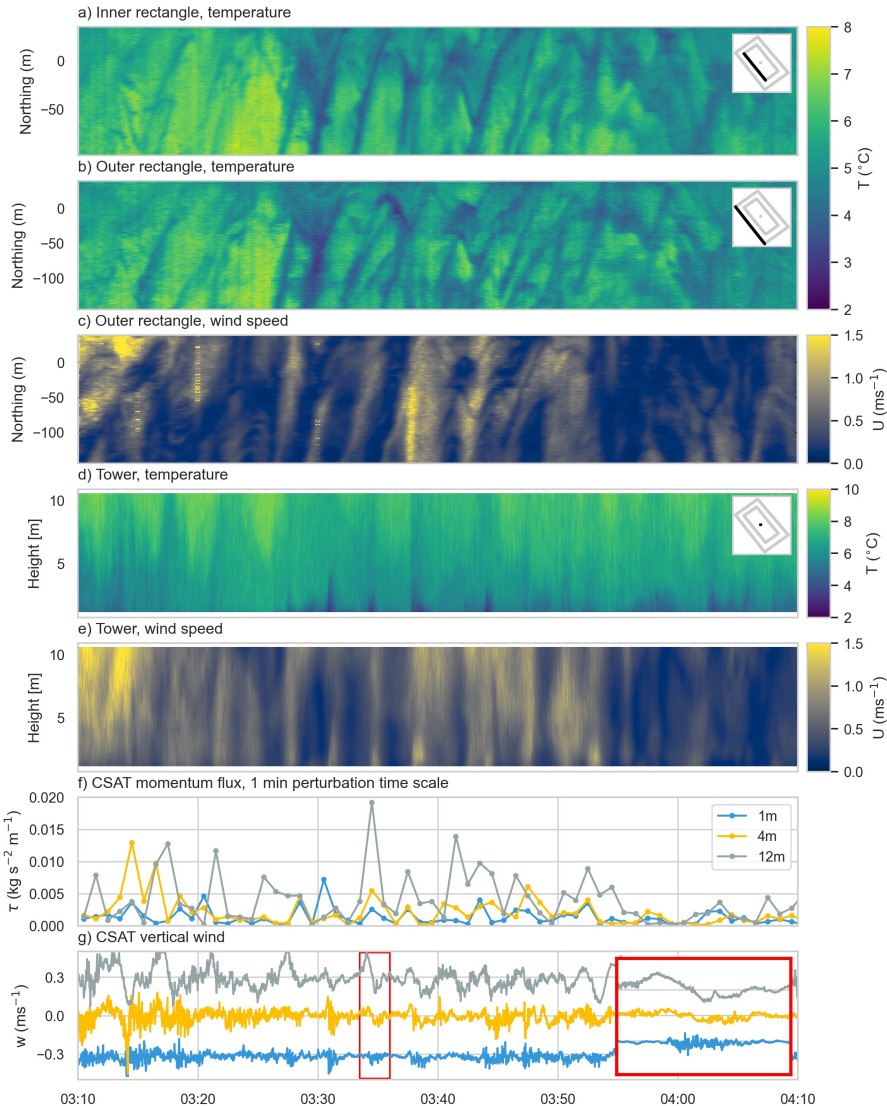

**Figure 10.** The tower and surface DTS array for a subset period of Fig. 9. The (a) inner and (b) outer rectangle temperature and (c) wind speed observed similar wave structures propagating horizontally across the surface. Vertically, the (d) FODS tower temperature and (e) wind speed capture the vertical cross-section of these waves. (f) CSAT observations of (f) the momentum flux were calculated using a 1 min perturbation. (g) CSAT vertical wind speed at 20Hz suggest that these waves became distorted by near-surface stability, thereby generating turbulence. A subset (red rectangle) of the vertical wind speed in (g) is highlighted in the inset axes. The mean vertical wind speed is offset by $0.3 \, \mathrm{ms^{-1}}$ for each vertical level for clarity.

$0.67 \mathrm{ms^{-1}}$ for a structure 80 m wide. This advective speed was slower than the horizontal wind speeds observed at the tower

(1.0 to 1.6 ms$^{-1}$), consistent with other studies that have tracked submeso-scale structures (Mahrt et al., 2009; Pfister et al.,
2021b; Zeeman et al., 2015).

The cold portion of the submeso-scale structure interacted with the tower between 3:31-3:32 followed by the warm portion
between 3:32-3:33. When the cold portion of the submeso-scale structure interacted with tower, a burst of turbulence occurred
and was recorded by the sonic anemometers at 0.5m, 1.25m, and 4m. The 12 m sonic anemometer at the top of the tower did
not observe this burst of turbulence, suggesting that the effect of the submeso-scale structure on generating turbulence was
constrained to near the surface, consistent with its apparent depth. Additionally, all sonic anemometers observed a meandering
in the horizontal wind direction as the submeso-scale structure passed the tower, with the lowest sonic anemometer leading and
the higher sonic anemometers increasingly lagging with height.

Prior to the submeso-scale structure's arrival, the tower FODS observations revealed a decoupled boundary layer consistent
with the hockey-stick transition concept (e.g., Sun et al., 2012, 2020) with strong static stability and decreased wind speed
below 4 m and weaker static stability and stronger wind speeds above (Fig. 11k,m). Simultaneous with the cold portion of
the submeso-scale structure passing the tower, the wind speed accelerated between heights of 1 m to 4 m, forming a distinct
nose (Fig. 11o). Between 2 m and 4 m agl the stability decreased until the entire profile above 2 m was only weakly stable.
After the passage of the submeso-scale structure, the overall wind shear within observed by the tower FODS decreased and
the air across the entire profile cooled by approximately 2 K. The 4 m and 12 m sonic anemometers reported positive (upward)
sensible heat fluxes (Fig. 11o). While these fluxes would normally be considered non-physical, the temperature profiles from
the tower FODS demonstrated that isolated and short-lived negative, i.e., unstable temperature gradients, existed during this
period (Fig. 11k). This example is consistent with the results from Fritz et al. (2021), that the FODS observations have fine
enough resolution to capture rapid static stability changes and reversals that other methods cannot.

The FODS data from the LOVE19 provide a unique and powerful opportunity for exploring the nature of these types of
submeso structures and their influence on boundary layer dynamics. Specifically, this example demonstrates how turbulence
in the presence of submeso structures can be driven "from the side" as opposed to top-down or bottom-up factors typically
assumed (e.g., Van de Wiel et al., 2017; Sun et al., 2012). Additionally, the intermittent turbulence event was explicitly not as-
sociated with classically-defined turbulent eddies. This example further highlights the importance and novelty of the distributed
observations from LOVE19 for understanding submeso-turbulence interactions.

## 7 Recommendations for future FODS deployments

As there has been only one previous deployment of electrically-heated fibers in an atmospheric setting (Sayde et al., 2015), we
provide some general guidelines here for future deployments, both related to heating the fiber specifically and for atmospheric
DTS deployments more generally. The thin PE coating on the stainless steel fibers does not provide full electrical insulation.
Unfortunately, thin PE coatings are necessary to enable the DTS to have a fast enough response time to observe turbulent per-
turbations (Thomas and Selker, 2021). Consequently, special care must be taken with this technique. Even with the PE coating,
a small current was induced in the reference sections that degraded the performance of the reference PT-100s, especially in our

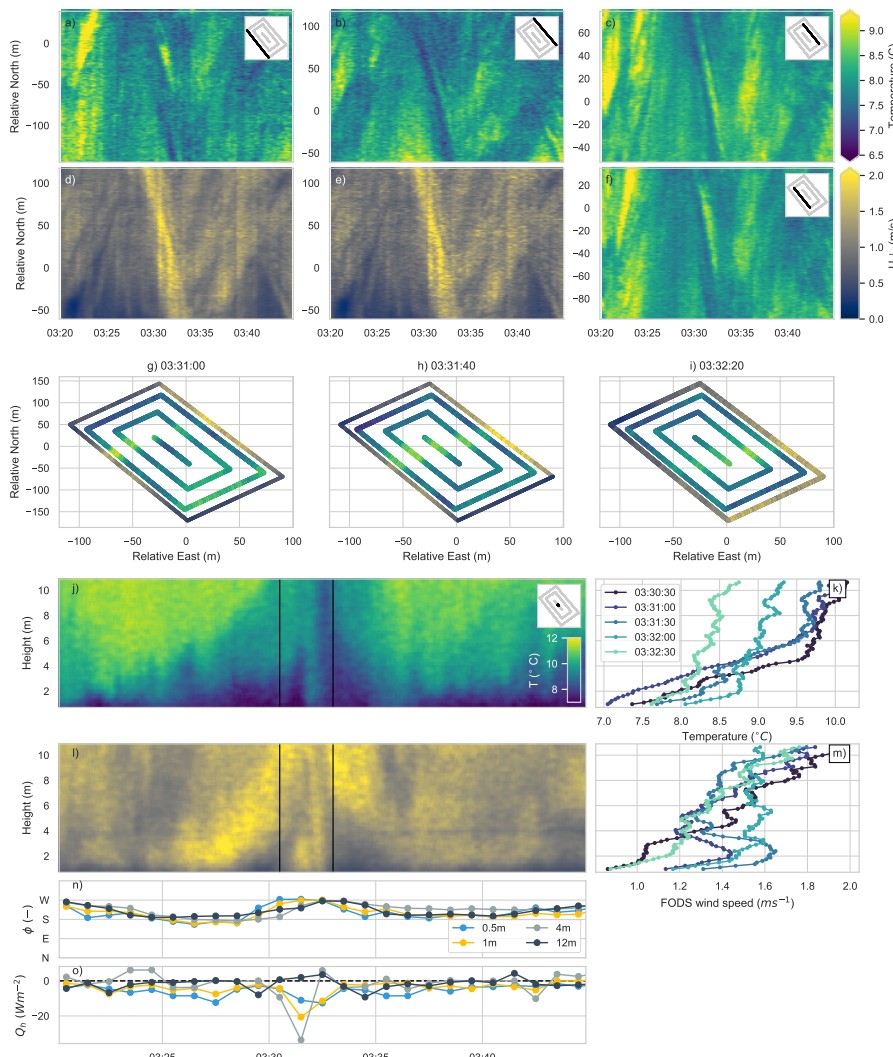

**Figure 11.** An example of a submeso-scale structure observed by the LOVE19 FODS array on July 22 2019. (a,b,c,f) DTS-observed temperatures from the outer and inner rectangle (see inset maps). (d-e) FODS wind speed along the outer rectangle. Three snapshots of the entire near-surface FODS array (g) as the leading cold portion of the submeso structure encountered the tower, (h) as the warm section encountered the tower, and (i) as the submeso structure propagated into the southeastern corner of the array. The tower observations of (j) FODS temperature and (l) FODS wind speed as the submeso structure passed the tower. The vertical lines indicate the period shown in (k, m) where each line is a 30s average vertical profiles. Finally, the CSAT observations of (n) horizontal wind direction and (m) sensible heat flux are shown.

initial testing with water reference baths as employed in most DTS experiments. It is necessary to electrically isolate the heated fibers from the reference sections, for instance by cutting the fiber and performing a splice of the optical core, which has an electrical resistance many orders of magnitude larger than the conducting stainless steel sheath. The downside of electrically

isolating the fiber using a splice is the introduction of step-losses which can degrade DTS calibration (van de Giesen et al., 2012; des Tombe et al., 2020).

Further, as seen in Fig. 3, power failures occurred during LOVE19. These were the result of the electrically-heated fibers grounding and triggering the false current protection of the site's electrical grid. It is imperative to electrically isolate fiber holders from ground and to keep vegetation away from the fiber, particularly during periods with frequent rainfall and dew
deposition. The thin PE coating, in combination with normal wear from an atmospheric deployment, made it so that power failures would occur when the wet grass contacted the heated fibers.

The fiber-optic cables were observed in a single-ended configuration. While the DTS performed reasonably within the validation reference sections and the amount of instrument white noise was acceptable, a problem emerged when comparing observations that should be identical, specifically the twisted pair of fibers used for the inner rectangle and the twisted pair of
555 fibers on FlyFOX. Spatially-dependent biases were found (not shown) even though the biases within the reference validation sections were small, but systematic. The most likely cause for the former was changes in the differential attenuation along the fiber e.g., caused by tension at the fiber holders (van de Giesen et al., 2012). The systematic biases in the validation reference sections are a result of combining the FODS cables into long optical path, which causes them to depend on one another as the backscatter light must pass through all cable sections. Single-ended calibration cannot account for these changes and artifacts.
As a result, we recommend that all future experiments employ a double-ended setup (van de Giesen et al., 2012; des Tombe et al., 2020), but saving raw backscatter Stokes and anti-Stokes data from both directions separately in a single-ended fashion as some information can be discarded in double-ended observation modes.

These results also suggest that future DTS work may benefit from including spatially-distributed evaluation of the DTS temperatures instead of evaluating the DTS temperatures at a small number of reference sections, which are often located
near the calibration reference sections. This behavior would not have been visible without the replicated temperatures from the twisted pair PVC fibers used for the inner rectangle. Additionally, even though DTS devices are capable of recording over 5 km of fiber length, we found that noise approximately doubled with every kilometer of fiber. Future deployments should consider the trade-off between decreasing the number of temporal samples through observing multiple channels and decreasing the instrument noise through observing shorter lengths of optical core on any given channel. Finally, the banding seen in Fig. 11
may be the result of using short integration times (i.e. the finest possible resolution from the respective DTS devices). Longer integration times would likely remove this issue at the expense of instrument's resolvable scales. Alternatively, longer reference sections may solve this issue.

Long-term monitoring intended to collect FODS observations over months and years will benefit from excellent electrical insulation of heated cable sections and temperature-stabilized environments for DTS instruments and reference sections. In en-
575 vironments experiencing large temperature swings, elongation and contraction of FODS cables made of stainless-steel sheaths need to be accounted and planned for in post-field calibration and when evaluating mechanical stresses from support elements. Sections of spare fiber between or within observational FODS elements allow for easier re-splicing after inevitable mechanical breaks without the need to rebuild larger FODS sections. Further recommendations can be found in Thomas and Selker (2021).

In the second example, the submeso-scale structure originated from outside the study area, potentially highlighting the need for even larger DTS arrays in future experiments. Previous research on the nature of submeso-scale structures using networks of point observations suggest that they exert influence across very large spatial scales up to kilometers (Abraham and Monahan, 2020; Pfister et al., 2021a). The LOVE19 dataset should provide insights into submeso structure-turbulence interactions, but the data may not be appropriate for studying their origins.

## 8 Conclusions

The Large-eddy Observatory - Voitsumra Experiment 2019 (LOVE19) for studying the weak-wind stable boundary layer (wwSBL) was presented. Understanding the wwSBL requires being able to separate motions on submeso scales from turbulence, as well as observations capable of resolving the spatiotemporal evolution of these motions (Mahrt and Thomas, 2016; Sun et al., 2015; Thomas, 2011; Zeeman et al., 2015; Pfister et al., 2021b, a). LOVE19 is able to fill this need by uniquely employing fiber-optic distributed sensing (FODS) of air temperature, wind speed, and wind direction. It is anticipated that LOVE19 will be of specific utility for the boundary-layer community, but also more broadly for communities studying the exchange of carbon, water vapor, and energy between the atmosphere and the surface. The uniqueness of the FODS arrays, in addition to the rich supporting boundary-layer observations, opens the door to addressing a wide range of research questions that could not be adequately addressed before.

## 9 Code and data availability

All data described in this manuscript are accessible through the DOI 10.5281/zenodo.4312976, including scripts for creating Fig. 9 and 11. The python package, pyfocs, used for creating the DTS data described here is available through the DOI 10.5281/zenodo.4292491. In addition the up-to-date pyfocs package can be found on github (https://github.com/klapo/pyfocs). The video supplement is available at the DOI 10.5446/53539

*Author contributions.* Conceptualization: KL, AFre, CT; Methodology: KL, AFre; Software: KL, AFre; Investigation: KL (lead), AFre, AFri, JS, JO, WB, CT; Formal Analysis: KL, CT; Data Curation: KL; Writing – Original Draft: KL; Visualization: KL; Supervision: KL and CT; Project Administration: CT; Funding Acquisition: CT

*Competing interests.* The authors declare no competing interests

*Acknowledgements.* This project has received funding from the European Research Council (ERC) under the European Union's Horizon 2020 research and innovation programme (grant agreement no. 724629 DarkMix), and from the Oberfranken Stiftung.

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
