# Peer review of "The Large-eddy Observatory - Voitsumra Experiment 2019 (LOVE19) with high-resolution, spatially-distributed observations of air temperature, wind speed, and wind direction from fiber-optic distributed sensing, towers, and ground-based remote sensing"

_Earth System Science Data, 2020_

## Referee Comment (RC2)

The manuscript describes a micrometeorological dataset consisting of observations from both traditional instrumentation and distributed fiber optic measurements of air temperature and wind speed. The authors describe the motivation behind the measurements, details on the instrumentation used, and present two examples of submeso-scale structures observed during the experiment.

The measurements are unique in the scales (hundreds of meters) and resolution (sub-meter) they cover, and will be very useful for micrometeorological research into the weak wind stable boundary layer.

While there are detailed explanations for most of the setup, some gaps are still present, mostly related to the DTS setup and calibration of the DTS data. Thus, I suggest that the authors consider the following comments and improve the manuscript.

**General comments**

It was a shame to see that the FODS wind direction data is not presented in this manuscript, even though this is in its title. I would like to ask the authors to add some data on this, if this is possible, even if it is just a comparison with the traditional sensors.

The calibration of the distributed temperature sensing data is glossed over quite quickly, with information on the setup missing, e.g., the temperatures and homogeneity of the reference sections, the length of fiber in the reference sections, which specific sections were used for calibration and which ones were used for validation.

In what kind of environment were the DTS machines located (temperature controlled or not?), and did their internal temperature show strong diurnal variations? I think that extra information on the setup could aid the reader with interpreting the results. A complete detailed analysis of the DTS calibration might take up too much space in this manuscript, and it possibly does not strongly affect the results of the wind speed measurements or submeso motion analysis. In that case I would like to advice the authors to skip figure 4 and just mention the mean biases in the text.

Lastly, how much uncertainty was introduced by the single-measurement calibration? If no time integration of calibration variables was done, the short (1s) integration time can make the temperature of the reference sections quite uncertain unless there is a long length of fiber in each reference section.

**Specific comments**

Line 47        'inelastic' does not add much information here, and might just confuse the reader.

Line 117/118 The use of spaces '1min' vs. '30 min' is inconsistent. Following the SI rule to add a space between the numerical value and unit symbol could make the text more clear (here and elsewhere).

Line 134        A verb is missing from "Air temperature and humidity"? E.g. 'were measured'

Line 156        If it is solid state it is not a 'bath'. Change this to e.g. 'solid state reference section'?

| Line 168 | Why did you not choose to employ a double-ended setup? Could you say more about the expected errors or uncertainty? |
|---|---|
| Line 171 | ΔLAF has not been explained previously |
| Line 187 | How were the heated and unheated cables attached to the tower, and at what distance from the tower itself? Could you elaborate on this here or in section 5.1 |
| Line 189 | Perhaps changing '50um' to the used fiber type (e.g. OM3) is more clear. The diameter of the internal waveguide does not help the reader much. (same for Line 200) |
| Line 200 | What does SBJ stand for? |
| Line 216 | What was the maximum range of the Ultima-DTS used? This was not specified earlier. |
| Line 218 | I do not agree with this conclusion based on the currently presented information. Too many other variables changed to make an assertion on the specific cause of the change in bias. At a minimum a more in-depth analysis of the calibration would be required to support this statement. |
| Line 226 | The optical fiber was loosely buffered. |
| Line 228 | Did the four optical cores differ so significantly in LAF that this was required? How much did they differ in length? |
| Line 230 | You state they were calibrated 'as' a single length of fiber. Was it one continuous fiber without any spices? This is not fully clear from the text.

If there were splices, calibrate over the splices? |
| Figure 4 | It is not clear to me which device was used where and when, or where along the fiber these validation sections were. |
| Line 251 | Could you specify 'OM3' (or OM4) instead of '50um'? |
| Line 251 | How were the water baths kept at a temperature, and how were they mixed? |
| Line 256 | It seems that the cable was spliced somewhere? This is not mentioned. Please add this and mention the issues of having a splice within a section of optical fiber. |
| Line 280 | Why would you not provide an already calculated wind speed for users of the data set to compare their calculations to? This would allow them to at least check if they made no large mistakes. |
| Line 300 | How did you treat the data in the case of the observed heated fiber temperature being lower than the non-heated fiber temperature? Due to, e.g., uncertainty in the DTS measurements. |
| Line 303 | Could you elaborate on the setup of the wind speed measurement along the tower? How were the cables attached, kept stable, etc. |

| | |
|---|---|
| Line 309 | How was the heating rate varied? Manually? Based on what information? |
| Line 309 | What does it mean to "optimize wind speed"? |
| Line 310 | What do you mean by "DTS bin"? |
| Line 313 | I am quite puzzled by this result; why would a lower heating rate lead to a bias? I would think that, in the case of perfect observations of fiber temperature, the bias would be very low at low heating rates and increase as the heating rate increases (due to generation of free convection). |
| | Is this a result of uncertainty in the temperature observations? What are other possible sources of the observed biases? Please elaborate. |
| Line 317 | Why are the biases compared to the heating rate, instead of the temperature difference? The temperature differences for each heating rate would vary according to the wind speed. Are the temperature differences not a cause of bias, as opposed to the heating rate. |
| Line 324 | Would these high temperature differences (up to 31 K!) lead to free convection? Did you check your data to see how dominant forced convection was, using e.g., the Archimedes number. |
| Line 328 | What was the orientation of the tower? Judging from the photograph it looks like the cables are located approximately on the north-east side of the tower. I would assume that if one would take care during deployment, the shortwave radiative shadow errors can be largely eliminated. |
| Line 335 | I am a bit disappointed to not see any results of wind direction presented here even if it was a simple comparison with the CSAT. Could you add such a short example here? |
| Line 338 | I assume you mean "…as FODS wind speed."? |
| Line 356 | The (lack of) wave-like patterns in the first ten minutes is not too clear to me from looking at the FODS wind speed in the figure (I would say that the waves start at the same time in both fig. 6e and 6g). It is quite difficult to see this in a 2D color plot. Were the wave-like patterns visible in the CSAT data or not? |
| Line 362 | Do you mean tower observations of wind speed? I do not see FODS wind direction presented here. |
| Line 378 | Is a video of this event available somewhere, e.g., as a supplement? That could make the motion of the event much more clear to the readers. |
| Line 405 | The 'upward-directed sensible heat fluxes' are not clear to me from the figure. Are negative sensible heat fluxes upward? I would expect the opposite, and I do not see the sign defined elsewhere |
| Line 420 | As you mention the response time here; do you have an estimate of response time for the different cables used? This would be very useful information for users analyzing the data. |

| Line 424 | If the fiber of the reference sections is spliced to the measure fiber this can degrade the calibration accuracy. Please mention this drawback. |
| | Would it not be possible to separate the metal tube from the optical fibers by breaking only the tube through, e.g., metal fatigue? This could provide electrical insulation without damaging or severing the optical fiber. |
| Line 426 | Was there a significant current flow through the wet grass, or were the power failures mainly the (essential!) safety features of the heat pulse system. Would low voltage heat controllers be a solution to this? Lower voltages (<50 V) would be much safer to handle and would have a much reduced current flow through the PE coating. |
| Line 436 | What would be the reason for saving the raw data separately in a single-ended fashion? What information is lost if you do not do this. |

**Technical corrections**

| Line 39 | Parentheses around Sun et al. |
| Line 42 | Here (eg. around 'DarkMix', and 'dark side') and elsewhere: the quotation marks only have the right hand version. ' ' vs. ' ' |
| Line 144 | Missing capitalization of Stokes and anti-Stokes |
| Line 152 | I think you meant to use 'i.e.' instead of 'e.g.' |
| Line 154 | Comma after 'coiled'. |
| Line 158 | Correct to 'thermoelectrically' |
| Line 182 | deltaLAF is in italic instead of regular font |
| Line 198 | "W m$^{-1}$" is in italic instead of regular font |
| Line 210 | Typo in 'kilometer' |
| Line 211 | 'inner and outer rectangle' instead of 'inner- and outer rectangle'. |
| Line 245 | "FLYFOX" capitalization is not consistent with "FlyFOX" elsewhere. |
| Line 291 | "W m$^{-1}$" is in italic instead of regular font |
| Line 331 | "ms$^{-1}$" is in italic instead of regular font |
| Line 354 | The time format in the text is inconsistent with the figures, could you homogenize this? |
| Line 392 | "…between 0311Z and 0322Z," |
| Line 397 | Extra "change"? Sentence is not fully clear. |
| Figure 7 | The label *k)* is not visible behind the legend. |

There is very little space between figure *k* and *m*, making the x-axis labels slightly unclear

Line 569    Remove "chap." from citation. Add a page number or remove "p."

---

## Author Comment (AC1)

**Reviewer 1:**
General Comments
* * *
This is a very interesting manuscript and presents an instrumentation array capable of resolving very fine spatially distributed scales of turbulent motion that I think will be of great interest to the community.

There is no doubt to me about the quality of work behind the manuscript and its importance.

The data is made available at the given repository and is well-organized.

The balance of the manuscript is generally good in terms of the attention paid to each section, though I would like to see the two example cases (second especially) fleshed out somewhat as the data presented is very interesting but difficult to follow.

Thank you for your constructive comments. We hope that we have adequately addressed them. Our responses are in blue.

As a general response, the example explanations have been simplified in order to emphasize just the features that are uniquely observable with FODS. The figure for the submeso soliton example has been revised to be clearer following reviewer comments. A movie of this example will also be provided in a revision to the data repository. The gravity wave example has been split into two components: the "column" observations from FlyFOX, SODAR-RASS, and LIDAR and the "surface" observations from the surface DTS array, tower DTS, and sonic anemometers (see Fig R2.6).

Specific Comments
* * *
Line 5 (style): per SI standard you should leave a space between units. E.g. '1350 m' instead of 1350m. This is applied inconsistently throughout the paper.
    This should now be fixed throughout the text.
25: Please clarify the statement that "[Taylor's Hyp. is] necessary to invoke when observing atmospheric turbulence". This is too reductive and it should be specified in which contexts this assumption is actually necessary.
    An example of when the hypothesis is necessary is now provided.
33: Consider here as well Mahrt (2008) Mesoscale wind direction shifts in the stable boundary-layer. Tellus 60A, 700-705
    Added.
42 (typographic): two sets of only closing apostrophes are used as opposed to an opening and a closing.
    Fixed.
46: the name of the experiment here is spelled differently than in the abstract.
    Fixed
49: "... has been shown to be accurately resolve air temperature" <-- grammar error
    Fixed.
75: 'demonstrates the unique observations from the [...] tethered balloon'. Please qualify this statement as this is not the first time that balloon-tethered DTS has been deployed, see Keller et al. (2011) in Atm. Meas. Techn. doi:10.5194/amt-4-143-2011
    We now emphasize how these observations are unique in section 4.3. The short story is that previous tethered DTS experiments were fundamentally limited by coarser resolutions, shorter flight durations, and/or lower flight heights. The FlyFOX-V observations were unique in their duration and resolution, as demonstrated by a recent study analyzing the morning transition with the FlyFOX-V observations (Fritz et al., 2021).

Fig. 1: this is quite minor but the labels d) and e) should be switched for readability as one follows a-c clockwise as opposed to row/column.
        Fixed.

As a general question relating to the temporal resolution: how were times synced and did the Ultima sample precisely at 1s? We found in field deployment that the actual temporal resolution was < 1 Hz though the data provided in the repository are only provided as seconds since x.
        Ultima and XT times were synced using an NTP client with a 50 microsecond tolerance. The Ultimas sampled for exactly a second (thankfully, see https://github.com/dtscalibration/python-dts-calibration/issues/106 for a discussion on the topic). However, the files are not generated every second, which is what leads to this frustrating < 1Hz resolution. The pyfocs processing code preserves the original time stamp through the calibration step. The final processing step, with the physically labeled coordinates, involves a linear interpolation to a uniform time stamp. These data with the uniform time stamp are in the Zenodo repository. The manuscript now includes the below sentence in section 4:
"Additionally, the DTS observations were interpolated to a uniform time step, as the actual time step constantly varied due to instrument idiosyncrasies."
127 (typographic): the s in Campbell Scientific should be capitalized
        Fixed
137: the soil measurements are never returned to. Are this mentioned simply for completeness?
        Yes, and they are available as part of the AWS netcdf in the Zenodo repository.
148: Silixa is located in Elstree
        Fixed
163: Could you provide an extra sentence or two on your reference baths: To which temperature was the cold-bath cooled? What was the step-regime for the warm bath? Was it warmed and then cooled continuously through some range? Which range? Etc.
        The requested information was added.
167/492: Your Hausner reference seems to be missing two authors: Selker and Tyler. Is the calibration process accounting for internal instrument temp?
        Fixed
179: Are you able to provide a quantitive estimate re: the accuracy of the alignment process and subsequently an induced error/confidence in the wind speed/dir measurements from the FODS array? This is quite important.

We agree it is quite important! We never followed through with a formal error analysis, which may be an interesting topic for a future technical note. We will, however, highlight that the alignment we conducted was sufficient as to reduce the errors to a negligible degree.

The alignment is accurate to within the DTS spatial resolution. This is verified a couple different ways, but the most convincing is that the distribution of temperature differences between the "heated" and "unheated" fibers during a period with no heating is normally distributed with a mean difference of 0.001 K (the histogram of temperature differences between the two fibers during a 26 hour period with no heating is shown in Fig R1.1). In other words, the two branches observed identical conditions as we would expect if they were correctly aligned. A bias emerged when they were not aligned correctly.

[Figure]

Figure R1.1. Histogram of the unheated and heated fibers along the 12m tower during a period with no heating. The mean difference was 0.001K and the differences are well described by a gaussian distribution consistent with differencing two noisy signals.

This alignment can be further exemplified using the output from the alignment tool for the contiguous 26 hour period without heating (Fig R1.2). The alignment was further verified in other periods without heating and for daytime versus nighttime (i.e. due to fiber artifacts changing) although we don't show this here. These tools are provided as part of the pyfocs software.

[Figure]

Figure R1.2. Comparing the unheated and "heated" fibers during a 26 hour long period with no heating. (a) The cross correlation between the time-averaged temperature profiles of the "heated" and unheated fibers clearly peaks at a shift of 0.381m. (b) This peak in the cross correlation is then used to interpolate

the heated fiber to a common coordinate with the unheated fiber, generating new LAF limits that correspond to the aligned section. The time averaged temperature profiles match, while the humps outside the region marked by the black lines correspond to the artifact from the fiber holders, which should not match due to different lengths of fiber wrapped around the holders. The shift is verified in the (c) spatial derivative (dT/dLAF) of the time averaged temperature profile. This process was repeated for sub-intervals with different meteorological conditions (e.g., day vs night and clouds vs no clouds). The optimal alignment did not vary.

182: dLAF is written with/without a space between delta and LAF and with/without italicized LAF. Please be consistent.
     Fixed
192: '1.3m height agl': height is tautological here.
     Fixed
198, 200, etc.: units are arbitrarily italicized or not throughout the manuscript, within the same sentence, etc.
     Fixed

203: the difference referred to here is due to the splice? The LAF? Cable-specific properties?
     I was deliberately vague here, as the real reason is perplexing and still unexplained. After some of the fiber holders the slope of log(Ps/Pas), log(Ps), and log(Pas) all changed, meaning the differential attenuation changed along the fiber. This change in slope cannot be a from a bend artifact, which can only generate step losses. Consequently, we are still unsure as to the physical cause for these changes in differential attenuation. The impact of these spatially and temporally persistent differences in differential attenuation can be seen in Fig. R1.3.

[Figure]

Fig. R1.3. The difference between the two pairs of twisted fibers around the inner rectangle for 6 h of data. The largest difference was 0.3 K and was stable in time.

Ideally, a doubled-ended calibration should fix this issue. As a result, using a single differential attenuation value for the entire fiber, as is the case for a single-ended calibration, leads to temperature differences within segments when comparing between twisted pair fibers (Fig R1.3). The error was never manifest in the reference sections, likely since the calibration forces the fiber temperatures to match there, hence our suggestion for more spatially distributed evaluations of DTS temperatures.

207: The abstract notes 1350 m of FODS measurements though there the cable length of 2.8 km is given. Perhaps you can clarify the lengths in a table or separate sentence(s) as the current reading is confusing.

Thank you for helping us catch a miscalculation (we discounted the heated length of fiber on the outer array and the coned fibers). The correct length of fiber is approximately 2105 m of fiber that was part observed as part of the array when including FlyFOX-V. However, you may note this is still below 2.8km. We expand upon this apparent discrepancy in the sections discussing the various elements of the FODS arrays.

The new text for the outer array reads:
"Due to this setup, in combination with excluding the second twisted pair of the inner rectangle, only 1.62~km of fiber optic cable of the available 2.8~km was retained as being part of the array."

And for the FODS-cross:
"A single optical core was 895m long. However, the fiber was routed between the study site and the reference sections (a distance of approximately 200m). This routing, in combination with the length of unused coned fiber and spare fiber kept at critical points in the array in case of a break, resulted in only observing 290 m of fiber optic core as part of the actual array out of the available 895 m."

210: "kilfometer".
Fixed.

Figure 4: Some additional context to these biases might be helpful. What are we to interpret from a slight over/under-estimation? Does this affect the wind measurements or only the temperature? Can you speak to the (presumably) diurnal cycle? Perhaps I've misunderstood something but how is a separate bias for the inner vs. Outer rectangle calculated if they're spliced to the same cable? Or does 'each cable' in line 214 mean 'each subdivision'? Please clarify.

We have generally revised the section regarding evaluations and the describing the fiber types to hopefully make the text clearer. Additionally, we have opted to remove any explicit guidance regarding a "good" or "bad" bias, instead opting to include an estimate of the noise contribution to the signal as a function of LAF. The new figure is included below. The method follows work by Peltola et al., (2021) in which the noise estimate was used to describe the time varying signal-to-noise ratio, thereby allowing users to filter for periods in which certain features may not be detectable depending on their needs. We also highlight that the validation bath always has a higher level of noise than the calibrated sections.

[Figure]

Fig R1.4. Noise as a function of LAF estimated following the Lenschow method (Lenschow et al., 2001) during a 26 h period without heating. The noise in the reference section is shown as the standard deviation of the reference sections during the evaluation period. Note that the Lenschow noise estimate and standard deviation of the reference sections strongly overlapped and so we only show the standard deviation for clarity. The Ultima data have been corrected by a factor of sqrt(10) to account for the difference in samples between the XT and Ultima due to differences in the temporal (5 s vs 1 s) and spatial (0.254 m vs 0.127 m) resolution. Even so, the Ultima still has a slightly noisier signal than the XT for a given LAF.

The noise contribution for the XT observations is quite small. The noise contribution for the Ultima observations is a bit larger than the noise from the XT, even after accounting for differences in the number of temporal and spatial samples between the two devices (Fig R1.4). This finding is generally consistent with our experience with these instruments across multiple field campaigns including those with double-ended calibration and longer integration times.

**Calibration evaluation and wind speed:**
The biases should not be relevant to the wind speed calculations along the tower – these fiber segments are within 20m LAF of each other. Even a very large differential attenuation bias would have a small impact on segments so near each other. Additionally, Fig R1.1 shows that the difference between the unheated and heated fibers was exceptionally small (0.001K). For the tower the largest problem was the uncertainty from subtracting two noisy signals.

247: no need to re-define agl
        Fixed.
248: stable boundary layer is uncapitalized here but capitalized elsewhere.
        Fixed.
252: length of reference sections?
        Added.

306: "The vertical orientation..." <- I'm confused by this sentence. Is angle of attack not considered because w ~= 0 near the surface or because of the vertical orientation of the cable? The sentence implies both.

The vertical orientation of the fiber and the negligible time averaged vertical wind speed are both necessary in order to neglect the angle of attack. If we put a tower with DTS on a slope and orientate it vertical with respect to gravity then these angle of attack justifications would no longer apply and one would need to include the angular effect of the along-slope flow. This section has been clarified to read: "For flows over flat surfaces, the near-surface flow deviates only a little from horizontal due to the small time-averaged vertical wind speed. Thus, the vertical orientation of the cables removes the angular dependence for deriving the horizontal wind speed since the angle of attack is orthogonal with respect to the fiber."

Figure 5 caption: "following van Ramshorts et al. van Ramshorts et al. (2020)"
        Fixed
355: I don't know what Fig. 1c is meant to illustrate to me in the context of this sentence
        We now specify that the fog can be seen in Fig 1c.
382: Labels should be Fig. 7a,b,c,f?
        Fixed.
386: Can you elaborate a bit on how you've determined the structure extent? Just by eye or some analytic approach?
        Answered below.
388: "at any given point the submeso-scale structure was present for approximately 2 minutes" how are you determining this? The structure bounds here are both the cold and warm section?
I determined the structure extent and the duration using continuous wavelet transforms in both time and space. During the event the spatial wavelet spectra peaked between 60-80m (as seen in the globally averaged wavelet power in Fig R1.5d). The duration that these enhanced scales were present at any given point corresponded to the structure's temporal duration. The analytic details, unfortunately, are beyond the scope of this data paper, but agree with what one would infer through visual inspection.

[Figure]

Fig R1.5: (a) The DTS temperature for a segment of the inner rectangle with the (b) spatial perturbation of temperature shown for clarity. Continuous wavelet transforms (CWT) were performed for each LAF bin across time (giving CWT in time) and for each time across space (giving CWT in space). The resulting cwt were averaged yielding (d). The submeso mode peaks at long spatial scales (> 60m). This enhancement was present at any given point for approximately two minutes.

Figure 7:

-> the colour maps are applicable to the entire plot?
Good catch! The tower needs its own color scale and now has one.

-> what are the limits of the wind dir plot? It's difficult to tell from this what the wind direction / direction of travel of the structure is
These have been added.

-> a vertical wind speed from the sonics is missing: does it remain ~0? How accurate is the w ~= 0 assumption employed by the FODS wind direction in this circumstance? What is estimated error associated?

We have added the vertical wind speed to a new figure (see Fig R2.5), partially to demonstrate that the vertical wind speed overall remains quite small in a time average. The exception is potentially on the upper segments of the cable where more coherent vertical oscillations were present. We now include a mention of anemometry studies (e.g., Ovink et al., 2001) which found that excluding the non-orthogonal wind components (i.e. the z-component in our case) leads to underestimating the velocity fluctuations, even when the non-orthogonal component was small. This discussion is included in the FODS wind speed section in which we discuss potential errors in the derivation of FODS wind speed.

-> what is the cause of the vertical banding in e)?
This is possibly a result of Reviewer 2's critique regarding the short integration. The banding in (e) is actually coming from the temperature data. It is present in the other subplots but is just less apparent visually. It is also noticeable as jumps in the bath temperatures as well. Constant, time invariant calibration parameters were also tested but were found to not be able to remove this banding.

-> are these plotted with interpolation=none? I was unable to find the script used to create these.
They were plotted with interpolation='nearest'. The scripts are called: `column_July18th_ESSD-example.ipynb` and ` horizontal-array_July22nd_ESSD-example.ipynb`.

-> There are 3 different date formats within this one plot
I actually counted 4. This has been amended to be a much more reasonable 2 time formats, one indicating time in minutes and one indicating time in seconds when a finer time scale is necessary.

**Reviewer 2**

The manuscript describes a micrometeorological dataset consisting of observations from both traditional instrumentation and distributed fiber optic measurements of air temperature and wind speed. The authors describe the motivation behind the measurements, details on the instrumentation used, and present two examples of submeso-scale structures observed during the experiment.

The measurements are unique in the scales (hundreds of meters) and resolution (sub- meter) they cover, and will be very useful for micrometeorological research into the weak wind stable boundary layer.

While there are detailed explanations for most of the setup, some gaps are still present, mostly related to the DTS setup and calibration of the DTS data. Thus, I suggest that the authors consider the following comments and improve the manuscript.

Thank you for your detailed feedback. We hope that we have adequately addressed your comments and strengthened the manuscript. Our responses are in blue.

**General comments**
It was a shame to see that the FODS wind direction data is not presented in this manuscript, even though this is in its title. I would like to ask the authors to add some data on this, if this is possible, even if it is just a comparison with the traditional sensors.
Unfortunately, the method was non-trivial to implement and develop. We have a paper describing the method in a "review race" with this one and I do not want to steal the thunder of the PhD student who put considerable effort into developing the method for environmental applications by including it here. We can only say to keep your eyes open for the upcoming paper!

The calibration of the distributed temperature sensing data is glossed over quite quickly, with information on the setup missing, e.g., the temperatures and homogeneity of the reference sections, the length of fiber in the reference sections, which specific sections were used for calibration and which ones were used for validation.

In what kind of environment were the DTS machines located (temperature controlled or not?), and did their internal temperature show strong diurnal variations? I think that extra information on the setup could aid the reader with interpreting the results. A complete detailed analysis of the DTS calibration might take up too much space in this manuscript, and it possibly does not strongly affect the results of the wind speed measurements or submeso motion analysis. In that case I would like to advice the authors to skip figure 4 and just mention the mean biases in the text.
Thank you for your thoughtful comments regarding the calibration.

We have expanded the explanation of the calibration to address your questions, specifically addressing the climate-controlled instrument trailer outside the study area and the properties of the reference sections requested.

Given the focus of both you and the other reviewer on uncertainty of the DTS, we opted to continue including Fig 4 with color coding to hopefully make the switching between DTS devices and the impact his has on calibration clearer. We additionally include an estimate of the instrument noise as a function of LAF (Fig R1.4). This method has been used by Peltola et al. (2021) to define a time varying signal-to-noise ratio, with which a user of the data could then filter out periods that cannot adequately resolve the feature(s) of interest. This addition also removes the judgement call in the manuscript on what is a "good" or "bad" calibration, providing information for future users to make their own decision.

Lastly, how much uncertainty was introduced by the single-measurement calibration? If no time integration of calibration variables was done, the short (1s) integration time can make the temperature of the reference sections quite uncertain unless there is a long length of fiber in each reference section. The "single-measurement" calibration using short time scales is the established method for atmospheric deployments as it is desirable to observe at as fine of a time scale as possible for studies of turbulence or near-turbulent time scales (e.g., Peltola et al., 2021, Thomas et al., 2012). The short integration time and short reference sections certainly increases the instrument uncertainty. We have added a measure of the white noise contribution (see Fig R1.4 and the response) as an attempt to communicate the instrument uncertainty.

However, it is possible that the short integration time and short reference sections may not contribute uncertainty that is only white noise. Figure 7 of the original manuscript includes "banding", which is highlighted in Fig R2.1. This banding appears simultaneously along the entire fiber, but is most apparent at longer LAFs, where presumably the signal-to-noise ratio has substantially decreased. The banding is not a white noise process. We now bring attention to this source of uncertainty in the text.

[Figure]

Fig R2.1. (a) temperature and (b) wind speed along the outer rectangle.

**Specific comments**

Line 47 'inelastic' does not add much information here, and might just confuse the reader.
Fixed.

Line 117/118 The use of spaces '1min' vs. '30 min' is inconsistent. Following the SI rule to add a space between the numerical value and unit symbol could make the text more clear (here and elsewhere).
Fixed.

Line 134 A verb is missing from "Air temperature and humidity"? E.g. 'were measured'
Fixed.

Line 156 If it is solid state it is not a 'bath'. Change this to e.g. 'solid state reference section'?
Fixed.

Line 168 Why did you not choose to employ a double-ended setup? Could you say more about the expected errors or uncertainty?
In hindsight this decision was likely an error. We thought that the multiple optical cores would provide an alternative path for reducing the noise of the DTS device. Please see the above discussion regarding the spatially estimated uncertainty.

Line 171 ΔLAF has not been explained previously

We now make it explicit that the use of the commas around ΔLAF are a renaming of the LAF bin size.

Line 187 How were the heated and unheated cables attached to the tower, and at what distance from the tower itself? Could you elaborate on this here or in section 5.1
Added to section 4.2

Line 189 Perhaps changing '50um' to the used fiber type (e.g. OM3) is more clear. The diameter of the internal waveguide does not help the reader much. (same for Line 200)
Here we are verbatim repeating the fiber's name from the supplier.

Line 200 What does SBJ stand for?
As above, we are regurgitating the manufacturer's cable name.

Line 216 What was the maximum range of the Ultima-DTS used? This was not specified earlier.
The Ultima variant is now specified.

Line 218 I do not agree with this conclusion based on the currently presented information. Too many other variables changed to make an assertion on the specific cause of the change in bias. At a minimum a more in-depth analysis of the calibration would be required to support this statement.
The FODS-cross (Fig 4c) was sampled by the higher-resolution Ultima until August 1$^{st}$, at which point everything was sampled by the lower-resolution, but ruggedized XT. The switch coincided with a reduction in the calibration bias. Between 16 July and 19 July all components were sampled by the Ultima, coinciding with the highest calibration biases for all components. On 20 July the PVC (Fig. 4a) and stainless-steel fibers (Fig. 4b) were then switched back to the XT, also coinciding with a reduction in the calibration biases. We hope the new version of the calibration figure makes this line of argument clearer.

[Figure]

Figure R2.2. New version of Fig 4, demonstrating the change in bias between the XT and Ultima using color to indicate which instrument was sampling which section of fiber. The biases and uncertainty were generally smaller for the XT compared to the Ultima.

Line 226 The optical fiber was loosely buffered.
     Fixed

Line 228 Did the four optical cores differ so significantly in LAF that this was required? How much did they differ in length?
Each optical core was spliced into one long optical cable. In hindsight this was a mistake, hence why we document it here. Each optical core was approximately 900m in length. Our estimates suggest that the noise doubles with every kilometer of cable for the Ultima. Given the already quite large noise floor and the approximate doubling of noise per km for this instrument (Fig R1.4) the data quality reduction was substantial.

Line 230 You state they were calibrated 'as' a single length of fiber. Was it one continuous fiber without any spices? This is not fully clear from the text. If there were splices, calibrate over the splices?
     We have completely revised this section to hopefully help with the clarity.

Figure 4 It is not clear to me which device was used where and when, or where along the fiber these validation sections were.

Please see Fig R2.2 above for a revision that makes this clearer. We also now highlight the location of the validation sections in Fig R1.4.

Line 251 Could you specify 'OM3' (or OM4) instead of '50um'?
Fixed.

Line 251 How were the water baths kept at a temperature, and how were they mixed?
We used an ambient temperature bath and a bath heated with an aquarium heater. Both were mixed with an aquarium pump. This information is now included with the FlyFOX-V description.

Line 256 It seems that the cable was spliced somewhere? This is not mentioned. Please add this and mention the issues of having a splice within a section of optical fiber.
This information is now discussed more clearly in the FlyFOX-V description.

Line 280 Why would you not provide an already calculated wind speed for users of the data set to compare their calculations to? This would allow them to at least check if they made no large mistakes.

Based on this comment and the following ones, we believe that we did not clearly communicate many aspects of FODS wind speed derivation and as a result we strongly revised the wind speed section. We hope that the new text is clearer.

We suspect that there are still unresolved issues in the wind speed calculation, such as the small convective heat flux from the unheated fiber in conditions when the unheated fiber is not in radiative equilibrium with the atmosphere. These potentially missing terms can be partially accounted for by treating the heating rate as a tunable parameter. By adjusting the heating rate as a tunable parameter, we could search for an optimal heating rate that would compensate for the potential errors in the wind speed calculation. The optimal heating rate did not agree with the observed heating rate, partially confirming that there are missing terms in the wind speed calculation. Further, the optimal heating rate varied with time of day and atmospheric conditions like cloudiness. As part of addressing this issue the text now includes the below paragraph:

"Initial analysis of FODS wind speed revealed time varying biases. In previous work with FODS wind speed, biases were attributed to angular effects. As angular effects could be neglected, it was suspected that instead there may be unresolved energy balance factors in the wind speed derivation. Regardless of the exact source of error, the heating rate can be treated as a tuning parameter that can be adjusted in order to compensate for these errors and thereby optimize FODS wind speed."

As for not including wind speed, size constraints played a large role in the decision. We strongly suspect that an improvement to the derivation of wind speed will become available in the future, so we prioritized the ability of future users of the data to re-calculate wind speed using the heated and unheated fibers. Additionally, we provide two examples of calculating the wind speed as part of the repository.

Unfortunately, a thorough consideration of the wind speed technique is beyond the scope of this data paper. Instead, our goal with this portion of the data paper was to highlight how there are unresolved terms in the wind speed calculation and that a user of the data must exercise caution when deriving this quantity.

Line 300 How did you treat the data in the case of the observed heated fiber temperature being lower than the non-heated fiber temperature? Due to, e.g., uncertainty in the DTS measurements.
This was never the case due to using a much higher heating rate than in any previous experiment. This point is now made clear in the text.

Line 303 Could you elaborate on the setup of the wind speed measurement along the tower? How were the cables attached, kept stable, etc.

Added to section 4.2.

Line 309 How was the heating rate varied? Manually? Based on what information?

The explanation has been updated to more clearly articulate that the heating rate was varied as part of a parameter search, not a physically varied heating rate.

Line 309 What does it mean to "optimize wind speed"?

See the comment above.

Line 310 What do you mean by "DTS bin"?

Changed to be "LAF bin"

Line 313 I am quite puzzled by this result; why would a lower heating rate lead to a bias? I would think that, in the case of perfect observations of fiber temperature, the bias would be very low at low heating rates and increase as the heating rate increases (due to generation of free convection).
Is this a result of uncertainty in the temperature observations? What are other possible sources of the observed biases? Please elaborate.

We refer the reviewer to Sayde et al. (2015) and van Ramshorst et al. (2020) as both studies go into detail on the underlying method. In both studies, the problem at higher wind speeds was that the faster wind speeds so strongly cooled the fiber that the temperatures of the heated and unheated fibers were no longer sufficiently different from each other (as implied in Reviewer 2's comment regarding line 300). In these cases a bias emerged. Higher heating rates mean that the heated fiber can experience a much stronger cooling convective heat flux from faster winds and still maintain a large enough temperature difference relative to the unheated fiber that one can still resolve the wind speed.

We now clearly define this error as the saturation effect in the text and (hopefully) explain it more clearly.

Line 317 Why are the biases compared to the heating rate, instead of the temperature difference? The temperature differences for each heating rate would vary according to the wind speed. Are the temperature differences not a cause of bias, as opposed to the heating rate.

We believe that the misunderstanding arose from our unclear explanation. The new explanation provided in the text should now make it clearer that the heating rate was not physically varied but held constant throughout the entire experiment.

Line 324 Would these high temperature differences (up to 31 K!) lead to free convection? Did you check your data to see how dominant forced convection was, using e.g., the Archimedes number.

Both vR20 and S15 consider the possibility of an unresolved heat flux from free convection. In the wind tunnel experiments of van Ramshorts et al. (2020), free convection is suspected to play a role in the disagreement between FODS and observed wind speeds at low wind speeds. While this may be the case in a wind tunnel, we did not find this to be the case in our environmental deployment in which turbulence is generally much more vigorous at all wind speeds. The effect of an unresolved forced convection term should be visible at low wind speeds, when the temperature difference between the heated cable and the air would be the largest. Instead, we find exceptionally good agreement at these low wind speeds, suggesting that the unresolved free convection term is safely neglected in a real atmospheric flow.

Line 328 What was the orientation of the tower? Judging from the photograph it looks like the cables are located approximately on the north-east side of the tower. I would assume that if one would take care during deployment, the shortwave radiative shadow errors can be largely eliminated.

While shading the fibers may be possible, it would require a large surface to block the sun throughout the entire day or the fiber would need to be rotated around the tower to stay in the shade. Unfortunately, neither is a realistic option. We do note that the shading of a forest canopy is a desirable side effect when observing in forested environments.

Line 335 I am a bit disappointed to not see any results of wind direction presented here even if it was a simple comparison with the CSAT. Could you add such a short example here?

As mentioned above, the derivation is significantly non-trivial. A manuscript demonstrating the FODS wind direction method is in a "review race" with this one. While we would love to show off the method, we must unfortunately refrain in this instance.

Line 338 The (lack of) wave-like patterns in the first ten minutes is not too clear to me from looking at the FODS wind speed in the figure (I would say that the waves start at the same time in both fig. 6e and 6g). It is quite difficult to see this in a 2D color plot. Were the wave-like patterns visible in the CSAT data or not?

In order to not entangle ourselves in a (very interesting!) discussion about the specifics of the gravity waves and thereby confuse the intent of the data paper, we have simplified the explanation of this example case. That said, we provide some new, further details. Unfortunately, the CSATs do not clearly elucidate this problem (Fig R2.3 and R2.4). I believe these would be characterized as "dirty gravity waves" (e.g., Cava et al., 2019; Sun et al., 2012):

[Figure]

Fig R2.3. Vertical wind observed at 20Hz by the CSAT data at the 12m, 4m, and 1.25m heights. The 0.5m CSAT is excluded due to poor instrument performance during this period, potentially due to water condensation on the transducer head.

[Figure]

Fig R2.4. Same as Fig R2.3 but focusing on a single wave event to demonstrate the "dirty" nature of the waves as observed by in the vertical wind field.

The 12m CSAT does show some very nice oscillations whereas nearer the surface there are instead bursts of vertical wind oscillations (e.g., FigR2.4), consistent with other work on dirty gravity waves becoming distorted by near surface stability (Cava et al., 2019; Sun et al., 2012). Unfortunately, a thorough analysis is beyond the scope of this data paper. This example is meant to, if you'll excuse the cheekiness, "exemplify" the potential utility of these novel data.

We also show the surface DTS array, which clearly indicates that the waves propagate along the surface between 3:10 and 3:40-50 from the south to the north, suggesting that these waves may sort of "roll down" the hill south of the site. After 3:50 or so the submeso modes becomes more dominant and the gravity wave oscillations in the temperature field become less apparent. The CSAT observed bursts of vertical wind coincide with the passage of the cold tongues of air from the gravity waves.

[Figure]

Fig R2.5 The surface DTS array along the inside rectangle clearly captures the propagation of internal gravity waves along the surface as shown in both the (a) temperature and the (b) the spatial perturbation of the temperature.

In order to communicate this information, we have split the gravity wave example into two pieces: a figure with the upper SBL observations and a figure with the surface data, as demonstrated in Fig R2.5.

[Figure]

Fig R2.6 New surface observations presented for the gravity wave example, demonstrating the "dirty gravity waves" propagating along the surface. The vertical wind in (g) has been offset by 0.3m/s for each level for visual clarity. The red rectangle in (g) corresponds to the inset axes outlined in red, which zooms in on a period of interest.

Line 356 I assume you mean "...as FODS wind speed."?
    We edited to sentence to hopefully be clearer.

Line 362 Do you mean tower observations of wind speed? I do not see FODS wind direction presented here.

> Fixed.

Line 378 Is a video of this event available somewhere, e.g., as a supplement? That could make the motion of the event much more clear to the readers

> A video will be included as part of the supplemental material.

Line 405 The 'upward-directed sensible heat fluxes' are not clear to me from the figure. Are negative sensible heat fluxes upward? I would expect the opposite, and I do not see the sign defined elsewhere

> We added a zero line for the sensible heat subplot in order to emphasize that the sensible heat flux switched sign to be directed upwards for short intervals coinciding with short lived unstable layers. Additionally, we now define the meaning of the sign of the sensible heat flux.

Line 420 As you mention the response time here; do you have an estimate of response time for the different cables used? This would be very useful information for users analyzing the data.

> Unfortunately, we do not have an estimate for each cable. The separation distance between the inner and outer rectangles and the CSATs precludes any meaningful analysis in that regard.

We do now include the estimate of the resolvable scales of the stainless steel along the tower observed by the Ultima (Fig R2.6). The resolvable scales of the DTS were determined using the spectral coherency between the sonic and DTS temperatures. The first frequency at which the coherency spectra drops under 0.01 is considered the time scale at which the DTS decorrelates with the sonic anemometer. This analysis is repeated with increasing spatial averages, yielding a curve from which the resolvable scales can be derived. The selection of the threshold is somewhat arbitrary, but yields a conservative estimate as significance testing for spectral coherence is not performative at small coherence values.

The stainless-steel fiber observed by the Ultima has a resolvable scale of approximately ~1.0 m and 10-11 s. Below these resolutions, the Ultima was not longer spectral coherent with the sonic temperature. Aggregations larger than these scales gain no additional spectral coherence. We highlight that the resolvable time scales presented here are all longer than those suggested Peltola et al. (2021), which found that the PVC fiber had a time resolvable scale of 2-3 s when using a short double-ended array and in Thomas et al., (2012) in which similarities in the spectra between sonic and DTS derived sensible heat flux were used to derive a resolvable time scale > 6.4 s. The differences may be due to different fibers and instruments, calibration procedures, or the method in determining the resolvable scale. Regardless, we argue that the coherence method has the advantage of not looking for similarity in the spectra, but rather resolving the time scales at which the DTS and reference sonics have covary spectrally as a function of spatial averaging. Consequently, this is the first direct estimate of the resolvable spatiotemporal scales from DTS in the atmosphere.

[Figure]

Fig R2.6. The resolvable time scales as a function of averaging length of the stainless-steel fiber on the tower at the heights indicated. A coherence threshold of 0.01 was chosen for indicating that the DTS observations were no longer spectrally coherent with the sonic temperatures.

Line 424 If the fiber of the reference sections is spliced to the measure fiber this can degrade the calibration accuracy. Please mention this drawback.
        Mentioned.
Would it not be possible to separate the metal tube from the optical fibers by breaking only the tube through, e.g., metal fatigue? This could provide electrical insulation without damaging or severing the optical fiber.
        It may be possible. The difficulty of breaking the metal sheath would be in removing the PE coating and then separating the sheath without severing the optical cable in the process. We found it difficult to splice this cable, requiring meticulous care for the optical fiber. The broken metal sheath often and easily severs the bare optical fiber. Breaking the sheath longitudinally (along the axis of the fiber instead of across it like for a splice) would be even more difficult. Your suggestion would be the ideal solution, but a user would need to weigh the time constraints against the small loss of signal from a splice, especially given the advancements in calibration techniques that can handle asymmetric step losses.

Line 426 Was there a significant current flow through the wet grass, or were the power failures mainly the (essential!) safety features of the heat pulse system. Would low voltage heat controllers be a solution to this? Lower voltages (<50 V) would be much safer to handle and would have a much reduced current flow through the PE coating.

Later experiences taught us that the false current protection is triggered when there is even a very small amount of voltage leaked from the heated array to the ground. Anything larger than 20-50microVolts created problems for us (the limit for the false current protectors on our power grids).

We suspect that the biggest issue to consider with low voltage heating would be the reduction in the heating rate. The heating rate often creates limitations for FODS, e.g. forming a ceiling for the highest FODS wind speed that can be observed. Low voltage heating may need a larger number of units heating smaller sections of fiber in which case care would need to be taken to carefully define the heating rate for each individual section. As long as a sufficient heating rate was provided and the heating rate was mapped to the fiber, the distributed low-voltage solution is certainly viable.

Line 436 What would be the reason for saving the raw data separately in a single- ended fashion? What information is lost if you do not do this.
For Silixa instruments, the LAF of the reverse channel is thrown out when observing in a double-ended mode. Additionally, it is easy to have errors in some parameters of the DTS configuration file making it so that the forward and reverse directions still need to be aligned after the fact anyway. Two-single ended observations preserve all information.

Technical corrections

Line 39 Parentheses around Sun et al.
Fixed.
Line 42 Here (eg. around 'DarkMix', and 'dark side') and elsewhere: the quotation marks only have the right hand version. ' ' vs. ' '
Fixed.
Line 144 Missing capitalization of Stokes and anti-Stokes
Fixed throughout the manuscript.
Line 152 I think you meant to use 'i.e.' instead of 'e.g.'
Fixed.
Line 154 Comma after 'coiled'.
Fixed.
Line 158 Correct to 'thermoelectrically'
Fixed.
Line 182 deltaLAF is in italic instead of regular font
Fixed.
Line 198 "W m$^{-1}$" is in italic instead of regular font
Fixed.
Line 210 Typo in 'kilometer'
Fixed.
Line 211 'inner and outer rectangle' instead of 'inner- and outer rectangle'.
Fixed.
Line 245 "FLYFOX" capitalization is not consistent with "FlyFOX" elsewhere.
Fixed throughout the text.
Line 291 "W m$^{-1}$" is in italic instead of regular font
Fixed throughout the text.
Line 331 "ms$^{-1}$" is in italic instead of regular font
Fixed.
Line 354 The time format in the text is inconsistent with the figures, could you homogenize this?
Fixed.
Line 392 "...between 0311Z and 0322Z,"

We thank the reviewer for the suggestion, but believe it is unnecessary in this location.
Line 397 Extra "change"? Sentence is not fully clear.

Fixed.

Figure 7 The label k) is not visible behind the legend.

Fixed

There is very little space between figure k and m, making the x-axis labels slightly unclear

Fixed

Line 569 Remove "chap." from citation. Add a page number or remove "p."

Fixed.

**Cited literature**

Fritz, A. M., Lapo, K., Freundorfer, A., Linhardt, T., & Thomas, C. K. (2021). Revealing the Morning Transition in the Mountain Boundary Layer using Fiber-Optic Distributed Temperature Sensing. *Geophysical Research Letters*, *2001*, 1–11. https://doi.org/10.1029/2020gl092238

Lenschow, D. H., Wulfmeyer, V., & Senff, C. (2000). Measuring Second- through Fourth-Order Moments in Noisy Data. *Journal of Atmospheric and Oceanic Technology*, *17*, 1330–1347. https://doi.org/https://doi.org/10.1175/1520-0426(2000)017<1330:MSTFOM>2.0.CO;2

Peltola, O., Lapo, K., Martinkauppi, I., O'Connor, E., K. Thomas, C., & Vesala, T. (2021). Suitability of fibre-optic distributed temperature sensing for revealing mixing processes and higher-order moments at the forest-air interface. *Atmospheric Measurement Techniques*, *14*(3), 2409–2427. https://doi.org/10.5194/amt-14-2409-2021

Cava, D., Mortarini, L., Anfossi, D., & Giostra, U. (2019). Interaction of Submeso Motions in the Antarctic Stable Boundary Layer. *Boundary-Layer Meteorology*, *171*, 151–173. https://doi.org/10.1007/s10546-019-00426-7

Thomas, C. K., Kennedy, A. M., Selker, J. S., Moretti, A., Schroth, M. H., Smoot, A. R., Tufillaro, N. B., & Zeeman, M. J. (2012). *High-Resolution Fibre-Optic Temperature Sensing : A New Tool to Study the Two-Dimensional Structure of Atmospheric Surface-Layer Flow*. 177–192. https://doi.org/10.1007/s10546-011-9672-7

Sun, J., Mahrt, L., Banta, R. M., & Pichugina, Y. L. (2012). Turbulence Regimes and Turbulence Intermittency in the Stable Boundary Layer during CASES-99. *Journal of the Atmospheric Sciences*, *69*(1), 338–351. https://doi.org/10.1175/JAS-D-11-082.1

Ovnik, R., Lamers, A. P. G. ., van Steenhoven, A. A., & Hoeijmakers, H. W. M. (2001). A method of correction for the binormal velocity fluctuation using the look-up inversion method for hot-wire anemometry. *Measurement Science and Technology*, *12*(8). https://doi.org/10.1088/0957-0233/12/8/330

---

## Author Response (AR1)

**List of Changes:**

- 1. Added an area overview map in response to requests from data users.
- 2. Minor text changes and clarifications throughout manuscript.
- 3. Revised the discussion of the DTS calibration.
  - a. Added characterization of the noise and associated figure.
  - b. Added characterization of the resolvable scales of observation with the DTS and associated figure.
  - c. Expanded and clarified description of the fiber geometry.
- 4. Revised description of FlyFOX-V, specifically including more details on fiber geometry, evaluation, and calibration.
- 5. Generally reorganized and clarified the text regarding the derivation and evaluation of FODS wind speed.
- 6. Simplified and streamlined the description of the two example use cases of the FODS data.
  - a. First example generalized to be about using FODS to study Internal Gravity Waves and includes surface observations from DTS and high-resolution CSAT data.
  - b. To accommodate these new observations, the figure for this example was split into two individual figures.
  - c. General improvements to near-surface submeso motion figure.
  - d. Added a supplemental video for the near-surface submeso motion example.
- 7. Some additional clarification in the recommendations section.

**Reviewer 1:**

**General Comments**

**-----**

This is a very interesting manuscript and presents an instrumentation array capable of resolving very fine spatially distributed scales of turbulent motion that I think will be of great interest to the community.

There is no doubt to me about the quality of work behind the manuscript and its importance.

The data is made available at the given repository and is well-organized.

The balance of the manuscript is generally good in terms of the attention paid to each section, though I would like to see the two example cases (second especially) fleshed out somewhat as the data presented is very interesting but difficult to follow.

Thank you for your constructive comments. We hope that we have adequately addressed them. Our responses are in blue and changes to the text are in red when indicating a substantial alteration or addition.

As a general response, the example explanations have been simplified in order to emphasize just the features that are uniquely observable with FODS. The figure for the submeso soliton example has been revised to be clearer following reviewer comments. A movie of this example has been provided as video supplement. The gravity wave example has been split into two components: the "column" observations from FlyFOX, SODAR-RASS, and LIDAR and the "surface" observations from the surface DTS array, tower DTS, and sonic anemometers (see Fig R2.6). Generally, we opted to stay away from any further analysis or interpretation as that is explicitly against the scope of the ESDD journal.

We additionally added an overview map of the surrounding landscape and topography, as this has been consistently requested by users of the data.

**Specific Comments**

Line 5 (style): per SI standard you should leave a space between units. E.g. '1350 m' instead of 1350m. This is applied inconsistently throughout the paper.

This should now be fixed throughout the text.

25: Please clarify the statement that "[Taylor's Hyp. is] necessary to invoke when observing atmospheric turbulence". This is too reductive and it should be specified in which contexts this assumption is actually necessary.

An example of when the hypothesis is necessary is now provided.

"...assumptions necessary to invoke when observing atmospheric turbulence, for instance Taylor's hypothesis of frozen turbulence when converting the average flux in time at a point to represent the area-averaged flux, are not valid"

33: Consider here as well Mahrt (2008) Mesoscale wind direction shifts in the stable boundary-layer. Tellus 60A. 700-705

Added.

- 42 (typographic): two sets of only closing apostrophes are used as opposed to an opening and a closing. Fixed.
- 46: the name of the experiment here is spelled differently than in the abstract. Fixed
- 49: "... has been shown to be accurately resolve air temperature" <-- grammar error Fixed.

75: 'demonstrates the unique observations from the [...] tethered balloon'. Please qualify this statement as this is not the first time that balloon-tethered DTS has been deployed, see Keller et al. (2011) in Atm. Meas. Techn. doi:10.5194/amt-4-143-2011

We now emphasize how these observations are unique in section 4.3. The short story is that previous tethered DTS experiments were fundamentally limited by coarser resolutions, shorter flight durations, and/or lower flight heights. The FlyFOX-V observations were unique in their duration and resolution, as demonstrated by a recent study analyzing the morning transition with the FlyFOX-V observations (Fritz et al., 2021). We reorganized the introduction a bit to make this point.

- 1. The specific line in question was necessary to adjust anyway in order to accommodate splitting of the previous figure into two parts and the addition of the surface FODS array. "The first example demonstrates the unique insights from FODS for studying internal gravity waves (section \ref{sect:column example})."
- 2. The description of FlyFOX-V compares to the previous examples with more detail to address the reviewer's concerns: "While the tethered FODS technique has been used previously for atmospheric profiling \citep[e.g.][]{Keller2011, Higgins2018}, these previous studies were fundamentally limited in duration (30~min and 17~h), spatial (0.25~m and 1.0~m), and/or temporal resolution (20~s and 5~min). FlyFOX-V removed these limitations through its fine resolution and longer flight times, successfully resolving the fine-scale structure of the morning transition for the first time \citep {Fritz2021}."

Fig. 1: this is quite minor but the labels d) and e) should be switched for readability as one follows a-c clockwise as opposed to row/column.

Fixed.

As a general question relating to the temporal resolution: how were times synced and did the Ultima sample precisely at 1s? We found in field deployment that the actual temporal resolution was < 1 Hz though the data provided in the repository are only provided as seconds since x.

Ultima and XT times were synced using an NTP client with a 50 microsecond tolerance. The Ultimas sampled for exactly a second (thankfully, see https://github.com/dtscalibration/python-dts-calibration/issues/106 for a discussion on the topic). However, the files are not generated every second, which is what leads to this frustrating

Figure R1.1. Histogram of the unheated and heated fibers along the 12m tower during a period with no heating. The mean difference was 0.001K and the differences are well described by a gaussian distribution consistent with differencing two noisy signals.

This alignment can be further exemplified using the output from the alignment tool for the contiguous 26 hour period without heating (Fig R1.2). The alignment was further verified in other periods without heating and for daytime versus nighttime (i.e. due to fiber artifacts changing) although we don't show this here. These tools are provided as part of the pyfocs software.

Figure R1.2. Comparing the unheated and "heated" fibers during a 26 hour long period with no heating. (a) The cross correlation between the time-averaged temperature profiles of the "heated" and unheated fibers clearly peaks at a shift of 0.381m. (b) This peak in the cross correlation is then used to interpolate the heated fiber to a common coordinate with the unheated fiber, generating new LAF limits that correspond to the aligned section. The time averaged temperature profiles match, while the humps outside the region marked by the black lines correspond to the artifact from the fiber holders, which should not match due to different lengths of fiber wrapped around the holders. The shift is verified in the (c) spatial derivative (dT/dLAF) of the time averaged temperature profile. This process was repeated for sub-intervals with different meteorological conditions (e.g., day vs night and clouds vs no clouds). The optimal alignment did not vary.

The paragraph in question now includes the line: The alignment process successfully reduced the bias to 0.001 K for the tower during a 26 hour period without heating (not shown).

182: dLAF is written with/without a space between delta and LAF and with/without italicized LAF. Please be consistent.

Fixed

192: '1.3m height agl': height is tautological here. Fixed 198, 200, etc.: units are arbitrarily italicized or not throughout the manuscript, within the same sentence, etc.

**Fixed**

203: the difference referred to here is due to the splice? The LAF? Cable-specific properties?

I was deliberately vague here, as the real reason is perplexing and still unexplained. After some of the fiber holders the slope of log(Ps/Pas), log(Ps), and log(Pas) all changed, meaning the differential attenuation changed along the fiber. This change in slope cannot be a from a bend artifact, which can only generate step losses. Consequently, we are still unsure as to the physical cause for these changes in differential attenuation. The impact of these spatially and temporally persistent differences in differential attenuation can be seen in Fig. R1.3.

---

## Referee Report (RR1)

**Review of "The Large-eddy Observatory - Voitsumra Experiment 2019 (LOVE19) with high-resolution, spatially-distributed observations of air temperature, wind speed, and wind direction from fiber-optic distributed sensing, towers, and ground-based remote sensing"**

**General comments**

Overall, the manuscript is well-structured and presents a comprehensive overview of the data and potential applications. The FlyFOX-V fine resolution data from 2m up to 200m stands out with its potential for novel insights on boundary layer processes.

**Specific comments**

1. Were there any rain events during the campaign? It would be useful to discuss/mention if there are effects of rain or variability in relative humidity in the accuracy of the different measurements.

2. Lines 227-229: Could the data from the cable length connecting to the trailer also be useful? Maybe also consider providing these data.

3. Figure 5: The biases appear to be systematic (positive in 5a, and negative in 5b and 5c). Elaborate why this is case. If the biases are systematic a correction should be possible.

4. Section 7: It could be useful to discuss conditions for longer-term monitoring with FODS.

**Technical comments**

1. Line 49: Replace "and" for ";".

2. Figure 5: I don't see the thin grey lines mentioned in the caption.

3. Line 435: Revise the sentence.

---

## Author Response (AR3)

Point-to-point responses to reviewer #3, 12-Aug-2021

Authors: We thank reviewer3 for carefully reading our manuscript and evaluating it as 'excellent and good'. Please find our point-to-point responses to your suggestions below printed in blue.

Specific comments

**S1**. Were there any rain events during the campaign? It would be useful to discuss/mention if there are effects of rain or variability in relative humidity in the accuracy of the different measurements.

→ CT: Good point, thanks for bringing this up. We included a discussion of the effects of liquid precipitation in three sections of the manuscript:

i) Caption of Figure 3 now reads: "Power failures occurred periodically due to electrical isolation issues heating the fiber optic cables sometimes co-occurring with rain events"

ii) Section 2, ln 97-100, we added: "The number of days with significant precipitation ($\geq$ 1~mm) was 5, 3, and 8 for the three phases, respectively. Significant rains primarily affected the data availability of the active-heating FODS elements during phase 2 as the moisture sometimes led to electric short-circuiting between the FODS cables and grass cover, which resulted in power loss due to the false-current protection."

iii) Section 4, ln 437-424, we added: "Liquid water intercepted by the FODS cables during significant rain events presents a temporary source of uncertainty for FODS wind speed as it affects the term $(T\_s - T\_f)$ until all water attached to both unheated and heated cables has evaporated. Heated cables dried quicker compared to unheated cables, and drying time was shorter during day than night. The exact drying time was not further evaluated, but was on the order of tens of minutes, during which users are advised to exercise caution when investigating FODS wind speed. High relative humidity without intercepted liquid water present on the FODS cables was not found to have an effect on FODS performance."

**S2.** Lines 227-229: Could the data from the cable length connecting to the trailer also be useful? Maybe also consider providing these data.

→ CT: Thank you for the suggestion, but the data from these FODS sections cannot be evaluated in any meaningful fashion since cables may be touching the ground, support structures, and/ or be coiled up in tight loops. These sections were excluded in mapping and the data archives for the mentioned reasons, as our focus was on the described FODS elements. We decided to not add this information to the text.

**S3**. Figure 5: The biases appear to be systematic (positive in 5a, and negative in 5b and 5c). Elaborate why this is case. If the biases are systematic a correction should be possible.

→ CT: Thank you for noting this observation. As already discussed in the text, the bias of the section shown in Fig. 5 is the validation reference section (warm section with largest LAF) not used for calibration. While the biases are systematic for the individual FODS elements (subfigures a, b, c) , a simple correction offsetting the bias applied to the entire section of the specific element did not improve the data certainty. On the contrary, applying the latter led to increased differences between inner and outer array for conditions when spatial temperatures are expected to be very small (e.g. high wind speeds at night, not shown in the manuscript). The main reasons for this is the single-ended configuration in combination with the combined

optical path of the inner and outer FODS array, and location-specific mechanical stress due to holders . Because of the physical configuration, the backscatter from the outer array always must pass through the FODS section of the inner array, making the biases depend on each other with one being positive, the other negative. We already included this explanation in Section 7 (Recommendations for future FODS deployments, ln 540 ff), but amended it. The reported biases are the optimal choice for the intended scientific objectives and a useful 'worst case' estimates. Note that the objectives of LOVE19 was to observe turbulence and submeso-scale motions with time scales much less than the presented 1h averages, and with temperature differences by far exceeding the magnitude of the biases. Hence, the reported biases do not prevent any physical interpretation of the FODS observations. We modified section 7 to now read (ln 555 to 559: "Spatially-dependent biases were found (not shown) even though the biases within the reference validation sections were small, but systematic. The most likely cause for the former was changes in the differential attenuation along the fiber e.g., caused by tension at the fiber holders \citep{VandeGiesen2012}. The systematic biases in the validation reference sections are a result of combining the FODS cables into long optical path, which causes them to depend on one another as the backscatter light must pass through all cable sections. Single-ended calibration cannot account for these changes and artifacts."

**S4**. Section 7: It could be useful to discuss conditions for longer-term monitoring with FODS.
→ CT: Very good point, we added the following paragraph (ln 573 - 578): "Long-term monitoring intended to collect FODS observations over months and years will benefit from excellent electrical insulation of heated cable sections and temperature-stabilized environments for DTS instruments and reference sections. In environments experiencing large temperature swings, elongation and contraction of FODS cables made of stainless-steel sheaths need to be accounted and planned for in post-field calibration and when evaluating mechanical stresses from support elements. Sections of spare fiber between or within observational FODS elements allow for easier re-splicing after inevitable mechanical breaks without the need to rebuild larger FODS sections. Further recommendations can be found in \citet{Thomas2021_book}."

Technical comments
T1. Line 49: Replace "and" for ";". → CT: We apologize, but we could not find where this replacement was suggested. The sentence reads "This technique relies on the temperature-dependent Raman-backscatter from laser pulses transmitted along a fiber optic cable in order to resolve temperature at a fine spatial (as fine as 0.127 m) and temporal (as fine as 1 s) resolution (Selker et al., 2006; Tyler et al., 2009)." We believe that this sentence structure is correct. Maybe line numbers are different.

T2. Figure 5: I don't see the thin grey lines mentioned in the caption. →CT: This was a remnant text from a previous figure version, we now replaced it with the correct caption of the revised figure. It now reads "The black and blue colors are provided as guidance for selecting lower and higher quality data, respectively."

T3. Line 435: Revise the sentence. → CT: We apologize, but we could not find the sentence to be revised.
Line 435 in the revised submission reads: "The FODS wind direction method is predicated on a similar argument as FODS wind direction."

Line 435 in the revised submission reads: "As a result, we recommend that all future experiments employ a double-ended setup (van de Giesen et al., 2012; des Tombe et al., 2020), but saving raw backscatter stokes and anti-stokes data from both directions separately in a single-ended fashion. "
We believe that both sentences are correct, please advise and direct us to the specific sentence/ flaws. Maybe line numbers are different.